# DiBS: Differentiable Bayesian Structure Learning

**Lars Lorch**
ETH Zurich
Zurich, Switzerland
`lars.lorch@inf.ethz.ch`

**Jonas Rothfuss**
ETH Zurich
Zurich, Switzerland
`jonas.rothfuss@inf.ethz.ch`

**Bernhard Schölkopf**
MPI for Intelligent Systems
Tübingen, Germany
`bs@tuebingen.mpg.de`

**Andreas Krause**
ETH Zurich
Zurich, Switzerland
`krausea@ethz.ch`

## Abstract

Bayesian structure learning allows inferring Bayesian network structure from data while reasoning about the epistemic uncertainty—a key element towards enabling active causal discovery and designing interventions in real world systems. In this work, we propose a general, fully *differentiable* framework for *Bayesian structure learning (DiBS)* that operates in the continuous space of a latent probabilistic graph representation. Contrary to existing work, DiBS is agnostic to the form of the local conditional distributions and allows for joint posterior inference of both the graph structure and the conditional distribution parameters. This makes our formulation directly applicable to posterior inference of complex Bayesian network models, e.g., with nonlinear dependencies encoded by neural networks. Using DiBS, we devise an efficient, general purpose variational inference method for approximating distributions over structural models. In evaluations on simulated and real-world data, our method significantly outperforms related approaches to joint posterior inference.[1]

## 1  Introduction

Discovering the statistical and causal dependencies that underlie the variables of a data-generating system is of central scientific interest. Bayesian networks (BNs) [1] and structural equation models are commonly used for this purpose [2–5]. Structure learning, the task of learning a BN from observations of its variables, is well-studied, but computationally very challenging due to the combinatorially large number of candidate graphs and the constraint of graph acyclicity.

While structure learning methods arrive at a single plausible graph or its Markov equivalence class (MEC), e.g., [6–9], *Bayesian structure learning* aims to infer a full posterior distribution over BNs given the observations. A distribution over structures allows quantifying the epistemic uncertainty and the degree of confidence in any given BN model, e.g., when the amount of data is small. Most importantly, downstream tasks such as experimental design and active causal discovery rely on a posterior distribution over BNs to quantify the information gain from specific interventions and uncover the causal structure in a small number of experiments [10–15].

A key challenge in Bayesian structure learning is working with a posterior over BNs—a distribution over the joint space of (discrete) directed acyclic graphs and (continuous) conditional distribution parameters. Most of the practically viable approaches to Bayesian structure learning revolve around

---

[1]Our Python `JAX` implementation of DiBS is available at: `https://github.com/larslorch/dibs`

35th Conference on Neural Information Processing Systems (NeurIPS 2021).

Markov chain Monte Carlo (MCMC) sampling in combinatorial spaces and bootstrapping of classical score and constraint-based structure learning methods, e.g., in causal discovery [10–15]. However, these methods marginalize out the parameters and thus require a closed form for the marginal likelihood of the observations given the graph to remain tractable. This limits inference to simple and by now well-studied linear Gaussian and categorical BN models [16–18] and makes it difficult to infer more expressive BNs that, e.g., model nonlinear relationships among the variables. Due to the discrete nature of these approaches, recent advances in approximate inference and gradient-based optimization could not yet be translated into similar performance improvements in Bayesian structure learning.

In this work, we propose a novel, *fully differentiable framework for Bayesian structure learning* (DiBS) that operates in the continuous space of a latent probabilistic graph representation. Contrary to existing work, our formulation is agnostic to the distributional form of the BN and allows for inference of the joint posterior over both the conditional distribution parameters and the graph structure. This makes our approach directly applicable to more flexible BN models where neither the marginal likelihood nor the maximum likelihood parameter estimate have a closed form. We instantiate DiBS with the particle variational inference method of Liu and Wang [19] and present a general purpose method for approximate Bayesian structure learning. In our experiments on synthetic and real-world data, DiBS outperforms all alternative approaches to joint posterior inference of graphs and parameters and when modeling nonlinear interactions among the variables, often by a significant margin. This allows us to narrow down plausible causal graphs with greater precision and make better predictions under interventions—an important stepping stone towards active causal discovery.

## 2 Background

**Bayesian networks**    A Bayesian network $(\mathbf{G}, \boldsymbol{\Theta})$ models the joint density $p(\mathbf{x})$ of a set of $d$ variables $\mathbf{x} = x_{1:d}$ using (1) a directed acyclic graph (DAG) $\mathbf{G}$ encoding the conditional independencies of $\mathbf{x}$ and (2) parameters $\boldsymbol{\Theta}$ defining the local conditional distributions of each variable given its parents in the DAG. When modeling $p(\mathbf{x})$ using a BN, each variable is assumed to be independent of its non-descendants given its parents, thus allowing for a compact factorization of the joint $p(\mathbf{x} \,|\, \mathbf{G}, \boldsymbol{\Theta})$ into a product of local conditional distributions for each variable and its parents in $\mathbf{G}$.

**Bayesian inference of BNs**    Given independent observations $\mathcal{D} = \{\mathbf{x}^{(1)}, \ldots, \mathbf{x}^{(N)}\}$, we consider the task of inferring a *full posterior* density over Bayesian networks that model the observations. Following Friedman and Koller [20], given a prior distribution over DAGs $p(\mathbf{G})$ and a prior over BN parameters $p(\boldsymbol{\Theta} \,|\, \mathbf{G})$, Bayes' Theorem yields the joint and marginal posterior distributions

$$p(\mathbf{G}, \boldsymbol{\Theta} \,|\, \mathcal{D}) \propto p(\mathbf{G})p(\boldsymbol{\Theta} \,|\, \mathbf{G})p(\mathcal{D} \,|\, \mathbf{G}, \boldsymbol{\Theta}) \,, \tag{1}$$

$$p(\mathbf{G} \,|\, \mathcal{D}) \propto p(\mathbf{G})p(\mathcal{D} \,|\, \mathbf{G}) \tag{2}$$

where $p(\mathcal{D} \,|\, \mathbf{G}) = \int p(\boldsymbol{\Theta} \,|\, \mathbf{G})p(\mathcal{D} \,|\, \mathbf{G}, \boldsymbol{\Theta})d\boldsymbol{\Theta}$ is the marginal likelihood. Thus, $p(\mathbf{G} \,|\, \mathcal{D})$ in (2) is only tractable in special conjugate cases where the integral over $\boldsymbol{\Theta}$ can be computed in closed form. The Bayesian formalism allows us to compute expectations of the form

$$\mathbb{E}_{p(\mathbf{G}, \boldsymbol{\Theta} \,|\, \mathcal{D})}\Big[f(\mathbf{G}, \boldsymbol{\Theta})\Big] \qquad \text{or} \qquad \mathbb{E}_{p(\mathbf{G} \,|\, \mathcal{D})}\Big[f(\mathbf{G})\Big] \tag{3}$$

for any function $f$ of interest. For instance, to perform Bayesian model averaging, we would use $f(\mathbf{G}, \boldsymbol{\Theta}) = p(\mathbf{x} \,|\, \mathbf{G}, \boldsymbol{\Theta})$ or $f(\mathbf{G}) = p(\mathbf{x} \,|\, \mathbf{G})$, respectively [21, 22]. In active learning of causal BN structures, a commonly used $f$ is the expected decrease in entropy of $\mathbf{G}$ after an intervention [10–12, 14, 15]. Inferring either posterior is computationally challenging because there are $\mathcal{O}(d!2^{\binom{d}{2}})$ possible DAGs with $d$ nodes [23]. Thus, computing the normalization constant $p(\mathcal{D})$ is generally intractable.

**Continuous characterization of acyclic graphs**    Orthogonal to the work on Bayesian inference, Zheng et al. [9] have recently proposed a differentiable characterization of acyclic graphs for structure learning. In this work, we adopt the formulation of Yu et al. [24], who show that a graph with adjacency matrix $\mathbf{G} \in \{0, 1\}^{d \times d}$ does not have any cycles if and only if $h(\mathbf{G}) = 0$, where

$$h(\mathbf{G}) := \mathrm{tr}\left[(\mathbf{I} + \tfrac{1}{d}\mathbf{G})^d\right] - d \,. \tag{4}$$

If $h(\mathbf{G}) > 0$, the function can be interpreted as quantifying the *cyclicity* or *non-DAG-ness* of $\mathbf{G}$. Follow-up work has leveraged this insight to model nonlinear relationships [24–27], time-series data [28], in the context of generative modeling [29, 30], for causal inference [31–33], and contributed to its theoretical understanding [34, 35]. So far, a connection to Bayesian structure learning has been missing.

## 3 Related Work

The existing literature on Bayesian Structure Learning predominantly focuses on inferring the marginal graph posterior $p(\mathbf{G} \,|\, \mathcal{D})$. Since this requires $p(\mathcal{D} \,|\, \mathbf{G})$ to be tractable, inference is limited to BNs with linear Gaussian or Categorical conditional distributions [16–18]. By contrast, the formulation we introduce overcomes this fundamental restriction by allowing for joint inference of the graph and the parameters, thereby facilitating the active (causal) discovery of more expressive BNs.

**MCMC** Sampling from the posterior over graphs is the most general approach to approximate Bayesian structure learning. Structure MCMC (MC³) [36, 37] performs Metropolis-Hastings in the space of DAGs by changing one edge at a time. Several works try to remedy its poor mixing behavior [38–40]. Alternatively, order MCMC draws samples in the smaller but still exponential space of node orders, which typically requires a hard limit on the maximum parent set size [20]. Attempts to correct for its unintended structural bias are themselves NP-hard to compute and/or limit the parent size [38, 41]. By performing variational inference in a continuous latent space, the method we propose circumvents such mixing issues and parent size limitations.

**Bootstrapping** The nonparametric DAG bootstrap [42] performs model averaging by bootstrapping $\mathcal{D}$, where each resampled data set is used to learn a single graph, e.g., using the GES or PC algorithms [6, 7]. The obtained set of DAGs approximates the posterior by weighting each unique graph by its unnormalized posterior probability. In simple cases, a closed-form maximum likelihood parameter estimate may be used to approximate the joint posterior [14], but only if $p(\mathcal{D} \,|\, \mathbf{G})$ is tractable in the first place.

**Exact methods** A few notable exceptions use dynamic programming to achieve exact marginal inference in time $O(d2^d)$, which is only feasible for $d \leq 20$ nodes [43, 44]. In special cases, e.g., for tree structures or known node orderings, exact inference can be performed more efficiently [45, 46].

## 4 A Fully Differentiable Framework for Bayesian Structure Learning

### 4.1 General Approach

With the goal of moving beyond the restrictive conjugate setups required by most discrete sampling methods for Bayesian structure learning, we propose to transfer the posterior inference task into the latent space of a probabilistic graph representation. Our resulting framework is *consistent* with the original Bayesian structure learning task in (3), enforces the *acyclicity* of $\mathbf{G}$ via the latent space prior, and provides the *score* of the continuous latent posterior, thus making general purpose inference methods applicable off-the-shelf.

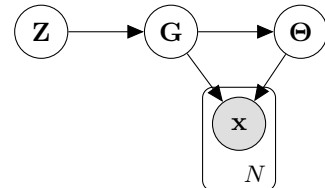

Figure 1: Generative model of BNs with latent variable $\mathbf{Z}$. This formulation generalizes the standard Bayesian setup in (1) where only $\mathbf{G}$, $\mathbf{\Theta}$, and $\mathbf{x}$ are modeled explicitly.

Without loss of generality, we assume that there exists a latent variable $\mathbf{Z}$ that models the generative process of $\mathbf{G}$. Specifically, the default generative model described in Section 2 is generalized into the following factorization:

$$p(\mathbf{Z}, \mathbf{G}, \mathbf{\Theta}, \mathcal{D}) = p(\mathbf{Z})p(\mathbf{G} \,|\, \mathbf{Z})p(\mathbf{\Theta} \,|\, \mathbf{G})p(\mathcal{D} \,|\, \mathbf{G}, \mathbf{\Theta}) \tag{5}$$

Figure 1 displays the corresponding graphical model. The following insight provides us with an equivalence between the expectation we ultimately want to approximate and an expectation over the posterior of the continuous variable $\mathbf{Z}$:

**Proposition 1** (Latent posterior expectation)**.** *Under the generative model in (5), it holds that*

$$(a) \quad \mathbb{E}_{p(\mathbf{G} \,|\, \mathcal{D})}\Big[f(\mathbf{G})\Big] = \mathbb{E}_{p(\mathbf{Z} \,|\, \mathcal{D})}\left[\frac{\mathbb{E}_{p(\mathbf{G} \,|\, \mathbf{Z})}\big[f(\mathbf{G})p(\mathcal{D} \,|\, \mathbf{G})\big]}{\mathbb{E}_{p(\mathbf{G} \,|\, \mathbf{Z})}\big[p(\mathcal{D} \,|\, \mathbf{G})\big]}\right] , \quad and$$

$$(b) \quad \mathbb{E}_{p(\mathbf{G}, \mathbf{\Theta} \,|\, \mathcal{D})}\Big[f(\mathbf{G}, \mathbf{\Theta})\Big] = \mathbb{E}_{p(\mathbf{Z}, \mathbf{\Theta} \,|\, \mathcal{D})}\left[\frac{\mathbb{E}_{p(\mathbf{G} \,|\, \mathbf{Z})}\big[f(\mathbf{G}, \mathbf{\Theta})p(\mathbf{\Theta} \,|\, \mathbf{G})p(\mathcal{D} \,|\, \mathbf{G}, \mathbf{\Theta})\big]}{\mathbb{E}_{p(\mathbf{G} \,|\, \mathbf{Z})}\big[p(\mathbf{\Theta} \,|\, \mathbf{G})p(\mathcal{D} \,|\, \mathbf{G}, \mathbf{\Theta})\big]}\right] .$$

A proof is provided in Appendix A.1. Rather than approximating $p(\mathbf{G} \,|\, \mathcal{D})$ or $p(\mathbf{G}, \mathbf{\Theta} \,|\, \mathcal{D})$, our goal will be to infer $p(\mathbf{Z} \,|\, \mathcal{D})$ or $p(\mathbf{Z}, \mathbf{\Theta} \,|\, \mathcal{D})$ instead, which by Proposition 1 allows us to compute

expectations of the form in (3). In the following, we first discuss how to define the two factors $p(\mathbf{G} \mid \mathbf{Z})$ and $p(\mathbf{Z})$ in a way that models only directed acyclic graphs. Then, we provide the details necessary to perform black box inference of the posteriors of the continuous latent variable $\mathbf{Z}$.

## 4.2 Representing DAGs in a Continuous Latent Space

**Generative model of directed graphs** We define the latent variable $\mathbf{Z}$ as consisting of two embedding matrices $\mathbf{U}, \mathbf{V} \in \mathbb{R}^{k \times d}$, i.e., $\mathbf{Z} = [\mathbf{U}, \mathbf{V}]$. Building on Kipf and Welling [47], we propose to use a bilinear generative model for the adjacency matrix $\mathbf{G} \in \{0, 1\}^{d \times d}$ of directed graphs using the inner product between the latent variables in $\mathbf{Z}$:

$$p_\alpha(\mathbf{G} \mid \mathbf{Z}) = \prod_{i=1}^{d} \prod_{j \neq i}^{d} p_\alpha(g_{ij} \mid \mathbf{u}_i, \mathbf{v}_j) \quad \text{with} \quad p_\alpha(g_{ij} = 1 \mid \mathbf{u}_i, \mathbf{v}_j) = \sigma_\alpha(\mathbf{u}_i^\top \mathbf{v}_j) \qquad (6)$$

where $\sigma_\alpha(x) = 1/(1 + \exp(-\alpha x))$ denotes the sigmoid function with inverse temperature $\alpha$. We denote the corresponding matrix of edge probabilities in $\mathbf{G}$ given $\mathbf{Z}$ by $\mathbf{G}_\alpha(\mathbf{Z}) \in [0, 1]^{d \times d}$ with

$$\mathbf{G}_\alpha(\mathbf{Z})_{ij} := p_\alpha(g_{ij} = 1 \mid \mathbf{u}_i, \mathbf{v}_j) \,. \qquad (7)$$

Since we model acyclic graphs, which do not contain self-loops, we set $p_\alpha(g_{ii} = 1 \mid \mathbf{Z}) := 0$. The latent dimensionality $k$ trades off the complexity of the variable interactions with tractability during inference. For $k \geq d$, the matrix of edge probabilities $\mathbf{G}_\alpha(\mathbf{Z})$ is not constrained in rank, and the generative model in (6) can represent any adjacency matrix without self-loops. That said, the size of the latent representation $\mathbf{Z}$ only grows $\mathcal{O}(d \cdot k)$ and, in principle, $k$ can be chosen independently of $d$. Note that the formulation in (6) models *directed* graphs since $\sigma_\alpha(\mathbf{u}_i^\top \mathbf{v}_j) \neq \sigma_\alpha(\mathbf{u}_j^\top \mathbf{v}_i)$. We can even interpret $\mathbf{u}_i$ and $\mathbf{v}_i$ as node embeddings that may encode more information than mere edge probabilities, e.g., for graph neural networks. The fact that $p_\alpha(\mathbf{G} \mid \mathbf{Z})$ is invariant to orthogonal transformations of $\mathbf{U}$ and $\mathbf{V}$ is not an issue in practice.

**Acyclicity via the latent prior distribution** A major constraint in learning BNs is the acyclicity of $\mathbf{G}$. With a latent graph model $p(\mathbf{G} \mid \mathbf{Z})$ in place, we design the prior $p(\mathbf{Z})$ to act as a soft constraint enforcing that only DAGs are modeled. Specifically, we define the prior of $\mathbf{Z}$ as the product of independent Gaussians with a Gibbs distribution that penalizes the *expected cyclicity* of $\mathbf{G}$ given $\mathbf{Z}$:

$$p_\beta(\mathbf{Z}) \propto \exp\left(-\beta \, \mathbb{E}_{p(\mathbf{G} \mid \mathbf{Z})}\left[h(\mathbf{G})\right]\right) \prod_{ij} \mathcal{N}(z_{ij}; 0, \sigma_z^2) \qquad (8)$$

Here, $h$ is the DAG constraint function given in (4) and $\beta$ is an inverse temperature parameter controlling how strongly the acyclicity is enforced. As $\beta \to \infty$, the support of $p_\beta(\mathbf{Z})$ reduces to all $\mathbf{Z}$ that only model valid DAGs. The Gaussian component ensures that the norm of $\mathbf{Z}$ is well-behaved. Traditional graph priors of the form $p(\mathbf{G}) \propto f(\mathbf{G})$ that induce, e.g., sparsity of $\mathbf{G}$, can be flexibly incorporated into $p_\beta(\mathbf{Z})$ by means of an additional factor involving $f(\mathbf{G}_\alpha(\mathbf{Z}))$ or $\mathbb{E}_{p(\mathbf{G} \mid \mathbf{Z})}[f(\mathbf{G})]$. Unless highlighting a specific point, we omit writing the hyperparameters $\alpha$ and $\beta$ to simplify notation.

## 4.3 Estimators for Gradient-Based Bayesian Inference

The extended generative model in (5) not only allows us to incorporate the notoriously difficult acyclicity constraint into the Bayesian framework. By rephrasing the posteriors $p(\mathbf{G} \mid \mathcal{D})$ and $p(\mathbf{G}, \mathbf{\Theta} \mid \mathcal{D})$ in terms of $p(\mathbf{Z} \mid D)$ and $p(\mathbf{Z}, \mathbf{\Theta} \mid D)$, respectively, it also makes Bayesian structure learning amenable to variational inference techniques that operate in continuous space and rely on the gradient of the unnormalized log posterior, also known as the *score*. The following result provides us with the score of both latent posteriors. Detailed derivations can be found in Appendix A.2.

**Proposition 2** (Latent posterior score). *Under the generative graph model defined in (5), the gradient of the log posterior density of $\mathbf{Z}$ is given by*

$$(a) \qquad \nabla_{\mathbf{Z}} \log p(\mathbf{Z} \mid \mathcal{D}) = \nabla_{\mathbf{Z}} \log p(\mathbf{Z}) + \frac{\nabla_{\mathbf{Z}} \, \mathbb{E}_{p(\mathbf{G} \mid \mathbf{Z})}\left[p(\mathcal{D} \mid \mathbf{G})\right]}{\mathbb{E}_{p(\mathbf{G} \mid \mathbf{Z})}\left[p(\mathcal{D} \mid \mathbf{G})\right]} \qquad (9)$$

*which is relevant when the marginal likelihood $p(\mathcal{D} \mid \mathbf{G})$ can be computed efficiently. In the general case, when inferring the joint posterior of $\mathbf{Z}$ and $\mathbf{\Theta}$, the gradients of the log posterior are given by*

$$(b) \qquad \nabla_{\mathbf{Z}} \log p(\mathbf{Z}, \mathbf{\Theta} \mid \mathcal{D}) = \nabla_{\mathbf{Z}} \log p(\mathbf{Z}) + \frac{\nabla_{\mathbf{Z}} \, \mathbb{E}_{p(\mathbf{G} \mid \mathbf{Z})}\left[p(\mathbf{\Theta}, \mathcal{D} \mid \mathbf{G})\right]}{\mathbb{E}_{p(\mathbf{G} \mid \mathbf{Z})}\left[p(\mathbf{\Theta}, \mathcal{D} \mid \mathbf{G})\right]} \qquad (10)$$

$$(c) \qquad \nabla_{\boldsymbol{\Theta}} \log p(\mathbf{Z}, \boldsymbol{\Theta} \mid \mathcal{D}) = \frac{\mathbb{E}_{p(\mathbf{G} \mid \mathbf{Z})}\big[\nabla_{\boldsymbol{\Theta}} p(\boldsymbol{\Theta}, \mathcal{D} \mid \mathbf{G})\big]}{\mathbb{E}_{p(\mathbf{G} \mid \mathbf{Z})}\big[p(\boldsymbol{\Theta}, \mathcal{D} \mid \mathbf{G})\big]} \qquad (11)$$

*where $p(\boldsymbol{\Theta}, \mathcal{D} \mid \mathbf{G}) = p(\boldsymbol{\Theta} \mid \mathbf{G})p(\mathcal{D} \mid \mathbf{G}, \boldsymbol{\Theta})$, which is efficient to compute by construction.*

The expectations have tractable Monte Carlo approximations because sampling from $p(\mathbf{G} \mid \mathbf{Z})$ in (6) is simple and parallelizable. The gradient terms of the form $\nabla_{\mathbf{Z}} \mathbb{E}_{p(\mathbf{G} \mid \mathbf{Z})}[\,\cdot\,]$, which also appear inside $\nabla_{\mathbf{Z}} \log p(\mathbf{Z})$, can be estimated using two different techniques, depending on the BN model inferred.

**Differentiable (marginal) likelihood**    Using the Gumbel-softmax trick [48, 49], we can separate the randomness from $\mathbf{Z}$ when sampling from $p(\mathbf{G} \mid \mathbf{Z})$ and obtain the following estimator:

$$\nabla_{\mathbf{Z}} \mathbb{E}_{p(\mathbf{G} \mid \mathbf{Z})}\big[p(\mathcal{D} \mid \mathbf{G})\big] \approx \mathbb{E}_{p(\mathbf{L})}\Big[\nabla_{\mathbf{G}}\, p(\mathcal{D} \mid \mathbf{G})\big|_{\mathbf{G}=\mathbf{G}_\tau(\mathbf{L},\mathbf{Z})} \cdot \nabla_{\mathbf{Z}}\, \mathbf{G}_\tau(\mathbf{L}, \mathbf{Z})\Big] \qquad (12)$$

where $\mathbf{L} \sim \text{Logistic}(0, 1)^{d \times d}$ i.i.d.. The matrix-valued function $\mathbf{G}_\tau(\cdot)$ is defined elementwise as

$$\mathbf{G}_\tau(\mathbf{L}, \mathbf{Z})_{ij} := \begin{cases} \sigma_\tau\left(l_{ij} + \alpha \mathbf{u}_i^\top \mathbf{v}_j\right) & \text{if } i \neq j \\ 0 & \text{if } i = j \end{cases} \qquad (13)$$

The estimator applies equally to $p(\boldsymbol{\Theta}, \mathcal{D} \mid \mathbf{G})$ in place of $p(\mathcal{D} \mid \mathbf{G})$. For the estimator to be well-defined, $p(\mathcal{D} \mid \mathbf{G})$ or $p(\boldsymbol{\Theta}, \mathcal{D} \mid \mathbf{G})$, respectively, needs to be differentiable with respect to $\mathbf{G}$. More specifically, the gradient $\nabla_{\mathbf{G}} p(\mathcal{D} \mid \mathbf{G})$ or $\nabla_{\mathbf{G}} p(\mathcal{D}, \boldsymbol{\Theta} \mid \mathbf{G})$ needs to be defined when $\mathbf{G}$ lies on the interior of $[0, 1]^{d \times d}$ and not at its discrete endpoints. This depends on the parameterization of the BN model we want to infer. In case $p(\mathcal{D} \mid \mathbf{G})$ or $p(\boldsymbol{\Theta}, \mathcal{D} \mid \mathbf{G})$ is only defined for discrete $\mathbf{G}$, it is possible to evaluate $\nabla_{\mathbf{G}} p(\mathcal{D} \mid \mathbf{G})$ or $\nabla_{\mathbf{G}} p(\mathcal{D}, \boldsymbol{\Theta} \mid \mathbf{G})$ using hard Gumbel-max samples of $\mathbf{G}$ (i.e., with $\tau = \infty$) and a straight-through gradient estimator. Since the DAG constraint $h$ is differentiable, the Gumbel-softmax trick can always be applied inside $\nabla_{\mathbf{Z}} \log p(\mathbf{Z})$. In practice, we always use $\tau = 1$.

**Non-differentiable (marginal) likelihood**    In general, $\nabla_{\mathbf{G}} p(\mathcal{D} \mid \mathbf{G})$ or $\nabla_{\mathbf{G}} p(\mathcal{D}, \boldsymbol{\Theta} \mid \mathbf{G})$ depending on the inference task might be not available or ill-defined. In this setting, the score function estimator provides us with a way to estimate the gradient we need [50]:

$$\nabla_{\mathbf{Z}} \mathbb{E}_{p(\mathbf{G} \mid \mathbf{Z})}\big[p(\mathcal{D} \mid \mathbf{G})\big] = \mathbb{E}_{p(\mathbf{G} \mid \mathbf{Z})}\Big[\big(p(\mathcal{D} \mid \mathbf{G}) - b\big)\, \nabla_{\mathbf{Z}} \log p(\mathbf{G} \mid \mathbf{Z})\Big] \qquad (14)$$

The estimator likewise applies for $p(\boldsymbol{\Theta}, \mathcal{D} \mid \mathbf{G})$ in place of $p(\mathcal{D} \mid \mathbf{G})$. Here, $b$ is a constant with respect to $\mathbf{G}$ that can be used for variance reduction [51], and $\nabla_{\mathbf{Z}} \log p(\mathbf{G} \mid \mathbf{Z})$ is trivial to compute. The derivations of both (12) and (14), alongside a more detailed discussion, can be found in Appendix B.

## 5    Particle Variational Inference for Structure Learning

In the previous section, we have proposed a *differentiable* formulation for *Bayesian structure learning (DiBS)* that is agnostic to the form of the local BN conditionals and, more importantly, translates learning discrete graph structures into an inference problem over the continuous variable $\mathbf{Z}$. In the following, we overcome the remaining challenge of inferring the intractable DiBS posteriors $p(\mathbf{Z} \mid \mathcal{D})$ and $p(\mathbf{Z}, \boldsymbol{\Theta} \mid \mathcal{D})$ by employing Stein variational gradient descent (SVGD) [19], a gradient-based and general purpose variational inference method. The resulting algorithm infers a particle approximation of the marginal or joint posterior density over BNs given observational data.

**SVGD for posterior inference**    Since Proposition 2 provides us with the gradient of the latent posterior score functions, we can apply SVGD off-the-shelf. SVGD minimizes the KL divergence to a target distribution by iteratively *transporting* a set of particles using a sequence of kernel-based transformation steps. We provide a more detailed overview of SVGD in Appendix C. Following this paradigm for DiBS, we iteratively update a fixed set $\{\mathbf{Z}^{(m)}\}_{m=1}^{M}$ or $\{\mathbf{Z}^{(m)}, \boldsymbol{\Theta}^{(m)}\}_{m=1}^{M}$ to approximate $p(\mathbf{Z} \mid \mathcal{D})$ or $p(\mathbf{Z}, \boldsymbol{\Theta} \mid \mathcal{D})$, respectively. If the BN model we aim to infer has a properly-defined likelihood gradient with respect to $\mathbf{G}$, we use the Gumbel-softmax estimator in (12) to approximate the posterior score. Otherwise, we resort to the score function estimator in (14). We use a simple kernel for SVGD:

$$k\big((\mathbf{Z}, \boldsymbol{\Theta}), (\mathbf{Z}', \boldsymbol{\Theta}')\big) := \exp\left(-\frac{1}{\gamma_z}\|\mathbf{Z} - \mathbf{Z}'\|_2^2\right) + \exp\left(-\frac{1}{\gamma_\theta}\|\boldsymbol{\Theta} - \boldsymbol{\Theta}'\|_2^2\right) \qquad (15)$$

with bandwidths $\gamma_z, \gamma_\theta$. For inference of $p(\mathbf{Z} \mid \mathcal{D})$, we leave out the second term involving $\boldsymbol{\Theta}, \boldsymbol{\Theta}'$. While $\mathbf{Z}$ is invariant to orthogonal transformations, more elaborate kernels that are, e.g., invariant to such transforms empirically perform worse in our experiments.

---

**Algorithm 1** DiBS with SVGD [19] for inference of $p(\mathbf{G} \mid \mathcal{D})$

---

**Input:** Initial set of latent particles $\{\mathbf{Z}_0^{(m)}\}_{m=1}^M$, kernel $k$, schedules for $\alpha_t, \beta_t$, and stepsizes $\eta_t$
**Output:** Set of discrete graph particles $\{\mathbf{G}^{(m)}\}_{m=1}^M$ approximating $p(\mathbf{G} \mid \mathcal{D})$

1: Incorporate prior belief of $p(\mathbf{G})$ into $p(\mathbf{Z})$                 ▷ See Section 4.2
2: **for** iteration $t = 0$ to $T - 1$ **do**
3:     Estimate score $\nabla_{\mathbf{Z}} \log p(\mathbf{Z} \mid \mathcal{D})$ given in (9) for each $\mathbf{Z}_t^{(m)}$     ▷ See (12) and (14)
4:     **for** particle $m = 1$ to $M$ **do**
5:         $\mathbf{Z}_{t+1}^{(m)} \leftarrow \mathbf{Z}_t^{(m)} + \eta_t\, \phi_t(\mathbf{Z}_t^{(m)})$                   ▷ SVGD step

      where $\phi_t(\cdot) := \dfrac{1}{M} \sum\limits_{k=1}^{M} \left[ k(\mathbf{Z}_t^{(k)}, \cdot)\, \nabla_{\mathbf{Z}_t^{(k)}} \log p(\mathbf{Z}_t^{(k)} \mid \mathcal{D}) + \nabla_{\mathbf{Z}_t^{(k)}} k(\mathbf{Z}_t^{(k)}, \cdot) \right]$

6: **return** $\{\mathbf{G}_\infty(\mathbf{Z}_T^{(m)})\}_{m=1}^M$                          ▷ See (16) and (17)

---

**Annealing $\alpha$ and $\beta$**    The latent variable $\mathbf{Z}$ not only probabilistically models the graph $\mathbf{G}$, but can also be viewed as a continuous relaxation of $\mathbf{G}$, with $\alpha$ trading off smoothness with accuracy. As $\alpha \to \infty$, the sigmoid $\sigma_\alpha(\cdot)$ converges to the unit step function. Hence, as $\alpha \to \infty$ in the graph model $p_\alpha(\mathbf{G} \mid \mathbf{Z})$ in (6), the expectations in Proposition 1 simplify to:

$$
\begin{aligned}
\mathbb{E}_{p(\mathbf{G} \mid \mathcal{D})}\Big[ f(\mathbf{G}) \Big] &\to \mathbb{E}_{p(\mathbf{Z} \mid \mathcal{D})}\Big[ f\big(\mathbf{G}_\infty(\mathbf{Z})\big) \Big] \\
\mathbb{E}_{p(\mathbf{G}, \boldsymbol{\Theta} \mid \mathcal{D})}\Big[ f(\mathbf{G}, \boldsymbol{\Theta}) \Big] &\to \mathbb{E}_{p(\mathbf{Z}, \boldsymbol{\Theta} \mid \mathcal{D})}\Big[ f\big(\mathbf{G}_\infty(\mathbf{Z}), \boldsymbol{\Theta}\big) \Big]
\end{aligned}
\tag{16}
$$

where $\mathbf{G}_\infty(\mathbf{Z})$ denotes the single limiting graph implied by $\mathbf{Z} = [\mathbf{U}, \mathbf{V}]$ and is defined as

$$
\mathbf{G}_\infty(\mathbf{Z})_{ij} := \begin{cases} 1 & \text{if } \mathbf{u}_i^\top \mathbf{v}_j > 0 \text{ and } i \neq j \\ 0 & \text{otherwise} \end{cases}
\tag{17}
$$

See Appendix A.3. In this case, $p_\alpha(\mathbf{G} \mid \mathbf{Z})$ converges to representing only a *single graph*. To be able to invoke this simplification, we anneal $\alpha \to \infty$ over the iterations of SVGD and, upon termination, convert the latent variables $\mathbf{Z}$ to the single discrete $\mathbf{G}_\infty(\mathbf{Z})$. Furthermore, we similarly let $\beta \to \infty$ in the latent prior $p_\beta(\mathbf{Z})$ over the iterations to enforce that the latent representation of $\mathbf{G}$ only models DAGs. By Equation (16), the resulting DAGs form a consistent particle approximation of $p(\mathbf{G} \mid \mathcal{D})$ or $p(\mathbf{G}, \boldsymbol{\Theta} \mid \mathcal{D})$, respectively. Algorithm 1 summarizes DiBS instantiated with SVGD for inference of $p(\mathbf{G} \mid \mathcal{D})$. The general case of inferring $p(\mathbf{G}, \boldsymbol{\Theta} \mid \mathcal{D})$ is given in Algorithm 2 of Appendix D.

**Single-particle approximation**    SVGD reduces to regular gradient ascent for the maximum a posteriori estimate when transporting only a *single* particle [19]. In this special case, DiBS with SVGD recovers some of the existing continuous structure learning methods: gradient ascent on a linear Gaussian likelihood solves an optimization problem similar to NOTEARS [9]. The cyclicity penalizer acts analogously. However, not only does DiBS automatically turn into a full Bayesian approach when using more particles, it is also not limited to settings such as linear Gaussian conditionals, where the adjacency matrix $\mathbf{G}$ and the parameters $\boldsymbol{\Theta}$ can be modeled together by a weighted adjacency matrix.

**Weighted particle mixture**    In high dimensional settings, it may be beneficial to move beyond a uniform weighting of the inferred particles of BN models to approximate $p(\mathbf{G} \mid \mathcal{D})$ or $p(\mathbf{G}, \boldsymbol{\Theta} \mid \mathcal{D})$. We consider a particle mixture that weights each particle by its unnormalized posterior probability $p(\mathbf{G}, \mathcal{D})$ or $p(\mathbf{G}, \boldsymbol{\Theta}, \mathcal{D})$, respectively, under the BN model. While we do not have a strong theoretical justification for the weighting, our motivation is that most of the particles in the empirical distribution will be unique as a consequence of the super-exponentially large space of DAGs, which may result in a crude approximation of their posterior probability mass function. In our experiments, DiBS and its instantiation with SVGD are used interchangeably, and DiBS+ denotes the weighted particle mixture.

## 6 Evaluation on Synthetic Data

### 6.1 Experimental Setup

**Synthetic data**    We compare DiBS to a set of related methods in marginal and joint posterior inference of synthetic linear and nonlinear Gaussian BNs. Our setup follows [9, 24, 27, 52], who consider inferring BNs with Erdős-Rényi and scale-free random structures [53, 54]. For each graph,

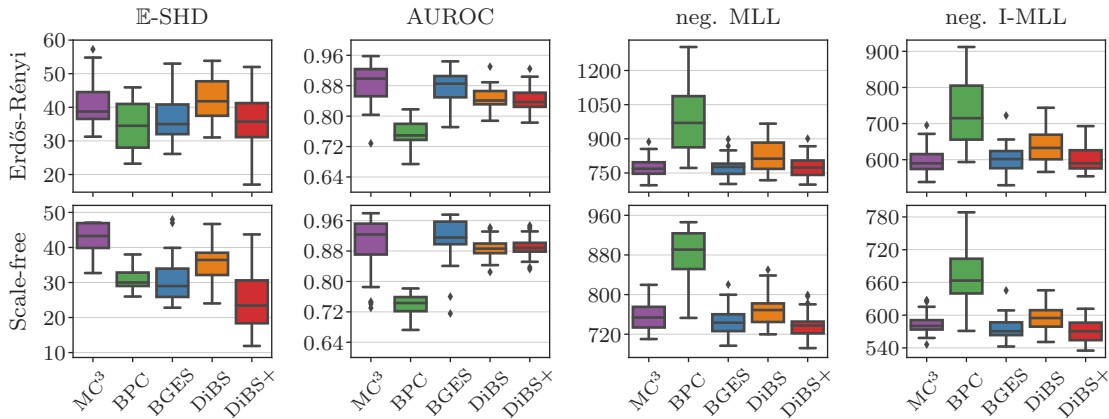

Figure 2: Marginal posterior inference of 20-node linear Gaussian BNs using the BGe marginal likelihood. Higher scores on AUROC and lower scores on $\mathbb{E}$-SHD, neg. MLL, neg. I-MLL are preferred. DiBS+ performs competitively across all metrics, in particular $\mathbb{E}$-SHD and neg. (I-)MLL.

here with $d \in \{20, 50\}$ nodes and $2d$ edges in expectation, we sample a set of ground truth parameters and then generate training, held-out, and interventional data sets. In all settings, we use $N = 100$ observations for inference, emulating the use case of Bayesian structure learning where the uncertainty about the graph structure is significant.

**Graph priors** For Erdős-Rényi graphs, all methods use the prior $p(\mathbf{G}) \propto q^{\|\mathbf{G}\|_1}(1-q)^{\binom{d}{2}-\|\mathbf{G}\|_1}$, capturing that each edge exists independently w.p. $q$ [53]. For scale-free graphs, we define the prior $p(\mathbf{G}) \propto \prod_{i=1}^{d}(1+\|\mathbf{G}_i^\top\|_1)^{-3}$, analogous to their power law degree distribution $p(\deg) \sim \deg^{-3}$ [54]. Here, $\mathbf{G}_i^\top$ is the $i$-th column of the adjacency matrix. DiBS implements either prior by using the corresponding term above as an additional factor in $p(\mathbf{Z})$ with $\mathbf{G} := \mathbf{G}_\alpha(\mathbf{Z})$ (see Section 4.2).

**Metrics** Since neither density nor samples of the ground truth posteriors are available for BNs of $d \in \{20, 50\}$ variables, we follow the evaluation metrics used by previous work. We define the *expected structural Hamming distance* to the ground truth graph $\mathbf{G}^*$ under the inferred posterior as

$$\mathbb{E}\text{-SHD}(p, \mathbf{G}^*) := \sum_{\mathbf{G}} p(\mathbf{G} \mid \mathcal{D}) \cdot \text{SHD}(\mathbf{G}, \mathbf{G}^*) \tag{18}$$

where $\text{SHD}(\mathbf{G}, \mathbf{G}^*)$ counts the edge changes that separate the essential graphs representing the MECs of $\mathbf{G}$ and $\mathbf{G}^*$ [8, 55]. In addition, we follow Friedman and Koller [20] and Ellis and Wong [41] and compute the *area under the receiver operating characteristic curve (AUROC)* for pairwise edge predictions when varying the confidence threshold under the inferred marginal $p(g_{ij} = 1 \mid \mathcal{D})$. Finally, following Murphy [10], we also evaluate the ability to predict future observations by computing the average *negative (marginal) log likelihood* on 100 held-out observations $\mathcal{D}^{\text{test}}$:

$$\text{neg. LL}(p, \mathcal{D}^{\text{test}}) := -\sum_{\mathbf{G}, \mathbf{\Theta}} p(\mathbf{G}, \mathbf{\Theta} \mid \mathcal{D}) \cdot \log p(\mathcal{D}^{\text{test}} \mid \mathbf{G}, \mathbf{\Theta}) \tag{19}$$

When inferring $p(\mathbf{G} \mid \mathcal{D})$, the corresponding neg. MLL metric instead uses $p(\mathcal{D}^{\text{test}} \mid \mathbf{G})$. Analogously, we also compute the *interventional* log likelihoods I-LL and I-MLL, a relevant metric in causal inference [10, 12]. Here, an interventional data set $(\mathcal{D}^{\text{int}}, \mathcal{I})$ is instead used to compute $p(\mathcal{D}^{\text{int}} \mid \mathbf{G}, \mathbf{\Theta}, \mathcal{I})$ and $p(\mathcal{D}^{\text{int}} \mid \mathbf{G}, \mathcal{I})$ in (19), respectively. Scores are the average of 10 interventional data sets. All reported metrics in this section are aggregated for inference of 30 random synthetic BNs.

In the remainder, DiBS is always run for 3,000 iterations and with $k = d$ for inference of $d$-variable BNs, which leaves the matrix of edge probabilities unconstrained in rank. We discard a DiBS particle in the rare case that a returned graph is cyclic. Complete details on Gaussian BNs, the evaluation metrics, hyperparameters, and all baselines can be found in Appendix E.

## 6.2 Linear Gaussian Bayesian Networks

**Marginal posterior inference** For linear Gaussian BNs, we first evaluate the classical setting of inferring the marginal posterior $p(\mathbf{G} \mid \mathcal{D})$ since $p(\mathcal{D} \mid \mathbf{G})$ can be computed in closed form. To this end, we employ the commonly used Bayesian Gaussian Equivalent (BGe) marginal likelihood, which

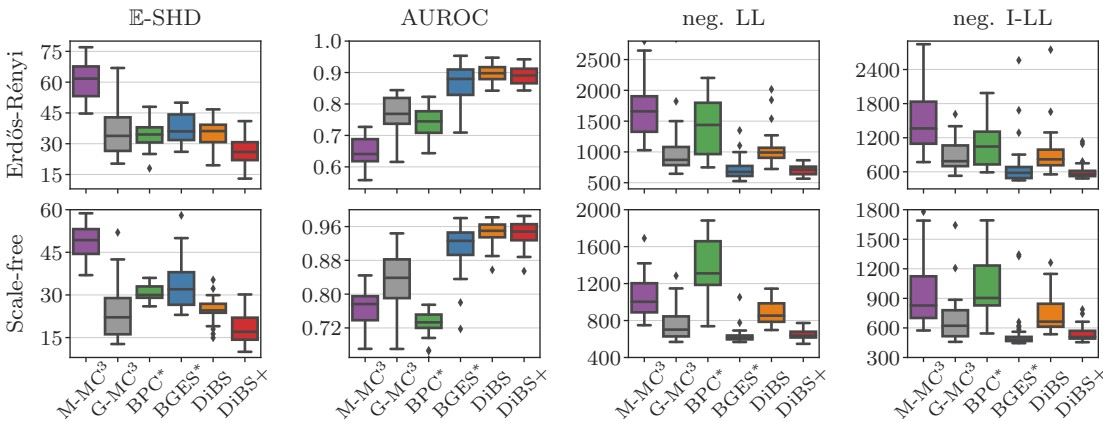

Figure 3: Joint posterior inference of graphs and parameters of linear Gaussian networks with $d = 20$ nodes. DiBS+ performs best across all of the metrics. BGES$^*$, the next-best alternative, yields substantially worse performance in $\mathbb{E}$-SHD, i.e., in recovering the overall graph structure and MEC. Recall that higher AUROC and lower $\mathbb{E}$-SHD, neg. LL, and neg. I-LL scores are preferable.

scores Markov equivalent structures equally [16, 17]. The form of the BGe score requires DiBS to use the score function estimator in (14).

We compare DiBS with the nonparametric DAG bootstrap [42] using the constraint-based PC [7] and the score-based GES [6] algorithms (BPC and BGES). For MCMC, we only consider structure MCMC (MC$^3$) [37] as a comparison. Order MCMC or hybrid DP approaches bound the number of parents and thus often exclude the ground truth graph a priori, especially for scale-free BN structures. Burn-in and thinning for MC$^3$ are chosen to make the wall time comparable with DiBS run on CPUs. In the remainder of the paper, each method uses 30 samples to approximate the posterior over BNs.

Figure 2 summarizes the results for 30 randomly generated BNs with $d = 20$ nodes. We find that DiBS+ performs well compared to the other methods, all of which were specifically developed for the marginal inference scenario evaluated here. DiBS+ appears to be preferable to DiBS.

**Joint posterior inference** When inferring the joint posterior $p(\mathbf{G}, \mathbf{\Theta} \,|\, \mathcal{D})$, we can employ a more explicit representation of linear Gaussian BNs, where the conditional distribution parameters are standard Gaussian. Here, DiBS can leverage the Gumbel-softmax estimator in (12) because $p(\mathbf{G}, \mathbf{\Theta} \,|\, \mathcal{D})$ is well-defined when $\mathbf{G}$ lies on the interior of $[0, 1]^{d \times d}$ (see Appendix E.1). To provide a comparison with DiBS in the absence of an applicable MCMC method, we propose two variants of MC$^3$ as baselines. Metropolis-Hastings MC$^3$ (M-MC$^3$) jointly samples parameters and structures, and Metropolis-within-Gibbs MC$^3$ (G-MC$^3$) alternates in proposing structure and parameters [56]. Moreover, we extend the bootstrap methods by taking the closed-form maximum likelihood estimate [57] as the posterior parameter sample for a given graph inferred using the BGe score (BPC$^*$ and BGES$^*$), an approach taken in, e.g., causal BN learning [14].

Figure 3 shows the results for $d = 20$ nodes, where $\mathbb{E}$-SHD and AUROC are computed by empirically marginalizing out the parameters. We find that DiBS+ is the only considered method that performs well across all of the metrics, often outperforming the baselines by a significant margin. As for marginal posterior inference of linear Gaussian BNs, DiBS+ performs slightly better than DiBS.

### 6.3 Nonlinear Gaussian Bayesian Networks

We also consider joint inference of *nonlinear* Gaussian BNs where the mean of each local conditional Gaussian is parameterized by a 2-layer dense neural network with five hidden nodes and ReLU activation functions (see Appendix E.1). Since the marginal likelihood does not have a closed form, we are unable to use BPC$^*$ and BGES$^*$ as a means of comparison. Figure 4 displays the results for $d = 20$ variables, where a given BN model has $|\mathbf{\Theta}| = 2{,}220$ weights and biases. Analogous to joint inference of linear Gaussian BNs, DiBS and DiBS+ outperform the MCMC baselines across the considered metrics. To the best of our knowledge, this is the first time that such nonlinear Gaussian BN models have been inferred under the Bayesian paradigm, which opens up exciting avenues in the active learning of more sophisticated causal structures.

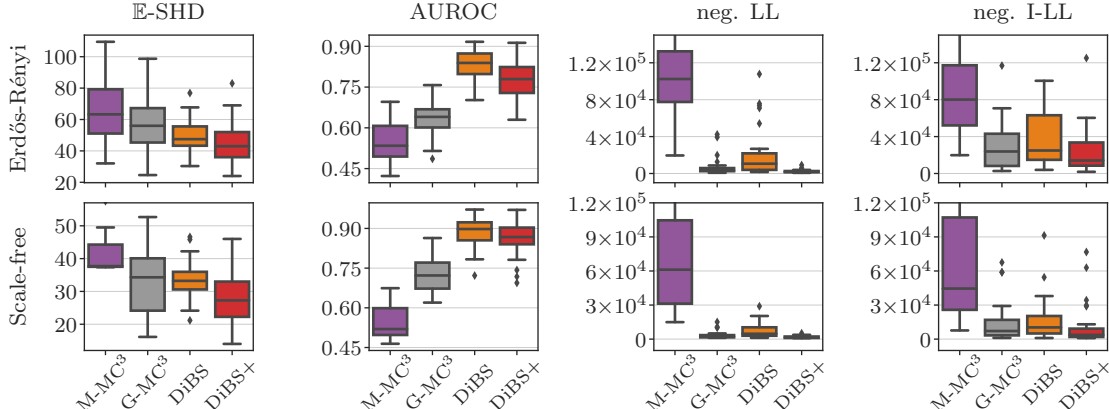

Figure 4: Joint posterior inference of nonlinear Gaussian BNs with $d = 20$ nodes. Here, the mean of the local conditional distribution of each node is parameterized by a 2-layer neural network with five hidden nodes. DiBS and DiBS+ perform favorably across the board, particularly in the graph metrics.

Appendix F complements Sections 6.2 and 6.3 with details on computing time and efficient implementation of DiBS with SVGD, showing that GPU wall times of the above inference experiments lie on the order of seconds or only a few minutes. In addition, Appendix G provides results for $d = 50$ variables, where DiBS+ likewise performs favorably when jointly inferring $p(\mathbf{G}, \mathbf{\Theta} \,|\, \mathcal{D})$. For marginal posterior inference of $p(\mathbf{G} \,|\, \mathcal{D})$ under the BGe marginal likelihood, DiBS appears to require more Monte Carlo samples to compensate for the high variance of the score function estimator in this high-dimensional setting.

### 6.4 DiBS with SVGD: Additional Analyses and Ablation Studies

Having compared DiBS and its instantiation with SVGD to existing approaches, we finally devote Appendix H to empirically analyzing some properties of the algorithm. One of our key results is that, all other things held equal, substituting our inner product model in (6) with $p_\alpha(g_{ij} = 1 \,|\, \mathbf{Z}) = \sigma_\alpha(z_{ij})$, where single *scalars* encode the edge probabilities, results in significantly worse evaluation metrics.

We additionally study the uncertainty quantification in (non)identifiable edge structures and show the effects of reducing the latent dimensionality $k$ or the number of iterations $T$. Our findings suggest that reducing either hyperparameter still allows for competitive posterior approximations and enables trading off posterior inference quality with computational efficiency, e.g., in large-scale applications.

## 7 Application: Inferring Protein Signaling Networks From Cell Data

A widely used benchmark in structure learning is the proteomics data set by Sachs et al. [3]. The data contain $N = 7,466$ continuous measurements of $d = 11$ proteins involved in human immune system cells as well as an established causal network of their signaling interactions.

We infer both linear and nonlinear Gaussian BNs with Erdős-Rényi graph priors exactly as in Section 6. The AUROC results in Table 1 indicate that the posterior by DiBS under the BGe model provides the most calibrated edge confidence scores. Not penalizing model complexity as much, the marginal BGe posterior of DiBS (DiBS+) averages a high expected number of 39.0 (35.0) edges, compared to 12.7 (14.2) and 12.6 (14.2) for its joint posteriors over linear and nonlinear BNs, respectively. Appendix I provides further details and analyses on this matter. The $\mathbb{E}$-SHD scores show that, among all the methods, DiBS is closest in structure to the consensus network when performing joint inference with nonlinear Gaussian BNs. This further highlights the need for nonlinear conditionals and joint inference of the graph and parameters in complex real-world settings.

Table 1: Inference of protein signaling pathways with Gaussian BNs. Metrics are the mean $\pm$ SD of 30 random restarts.

|  |  | $\mathbb{E}$-**SHD** | **AUROC** |
|---|---|---|---|
| † | MC³ | $34.0 \pm 0.7$ | $0.616 \pm 0.027$ |
|  | BPC | $25.5 \pm 2.3$ | $0.566 \pm 0.020$ |
|  | BGES | $33.7 \pm 1.7$ | $0.641 \pm 0.030$ |
|  | DiBS | $37.4 \pm 0.5$ | $\mathbf{0.647 \pm 0.047}$ |
|  | DiBS+ | $34.7 \pm 1.5$ | $0.629 \pm 0.045$ |
| § | M-MC³ | $37.3 \pm 3.5$ | $0.551 \pm 0.078$ |
|  | G-MC³ | $30.5 \pm 3.2$ | $0.527 \pm 0.067$ |
|  | DiBS | $23.4 \pm 0.5$ | $0.598 \pm 0.052$ |
|  | DiBS+ | $22.9 \pm 2.7$ | $0.557 \pm 0.052$ |
| ¶ | M-MC³ | $25.2 \pm 3.0$ | $0.526 \pm 0.084$ |
|  | G-MC³ | $35.1 \pm 3.2$ | $0.540 \pm 0.080$ |
|  | DiBS | $\mathbf{22.6 \pm 0.5}$ | $0.577 \pm 0.039$ |
|  | DiBS+ | $22.8 \pm 1.9$ | $0.535 \pm 0.041$ |

† Linear Gaussian BN; graph only via BGe marginal lik.
§ Linear Gaussian BN; graph and parameters jointly
¶ Nonlinear Gaussian BN; graph and parameters jointly

## 8 Conclusion

We have presented a general, fully differentiable approach to inference of posterior distributions over BNs. Our framework is based on a continuous latent representation of DAGs, whose posterior can be equivalently inferred—without loss of generality and using existing black box inference methods. While we have used SVGD [19] for this purpose, our general approach could also be instantiated with, e.g., gradient-based sampling methods that rely on the score of the target density [58, 59]. This may improve upon the asymptotic runtime of DiBS with SVGD, which scales quadratically in the number of sampled particles. We expect that our end-to-end approach can be extended to handle missing and interventional data as well as to amortized contexts, where rich unstructured data is available.

**Broader impact**   Our work is relevant to any scientific discipline that aims at inferring the (causal) structure of a system or reasoning about the effects of interventions. If algorithms and decisions are grounded in the structural understanding of a data-generating system and take into account the epistemic uncertainty, we expect them to be more robust and have fewer unforeseen side-effects. However, the assumptions allowing for a causal interpretation of DAGs, e.g., the absence of unmeasured confounders, are often untestable and to be taken with care [60], particularly in safety-critical and societally-sensitive applications. Hence, while potential misuse can never be ruled out, our presented method predominantly promises positive societal and scientific impact.

## Acknowledgments and Disclosure of Funding

This project received funding from the Swiss National Science Foundation under NCCR Automation under grant agreement 51NF40 180545, the European Research Council (ERC) under the European Union's Horizon 2020 research and innovation program grant agreement no. 815943, and was supported with compute resources by Oracle Cloud Services. This work was also supported by the German Federal Ministry of Education and Research (BMBF): Tübingen AI Center, FKZ: 01IS18039B, and by the Machine Learning Cluster of Excellence, EXC number 2064/1 – Project number 390727645. We thank Nicolo Ruggeri and Guillaume Wang for their valuable feedback.

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
