# A  Proofs of the Main Results

## A.1  Proposition 1

**Proof**  For ease of understanding, we recall that the generative model in (5) factorizes the joint distribution as $p(\mathbf{Z}, \mathbf{G}, \mathbf{\Theta}, \mathcal{D}) = p(\mathbf{Z})p(\mathbf{G} \mid \mathbf{Z})p(\mathbf{\Theta} \mid \mathbf{G})p(\mathcal{D} \mid \mathbf{G}, \mathbf{\Theta})$. First, let us consider case (a), the setting where the marginal likelihood $p(\mathcal{D} \mid \mathbf{G})$ can be computed in closed form. We get

$$\mathbb{E}_{p(\mathbf{G} \mid \mathcal{D})} f(\mathbf{G}) = \sum_{\mathbf{G}} p(\mathbf{G} \mid \mathcal{D}) f(\mathbf{G}) \tag{A.1}$$

$$= \sum_{\mathbf{G}} \frac{p(\mathbf{G}, \mathcal{D}) f(\mathbf{G})}{p(\mathcal{D})} \tag{A.2}$$

$$= \sum_{\mathbf{G}} \int_{\mathbf{Z}} \frac{p(\mathbf{Z}, \mathbf{G}, \mathcal{D}) f(\mathbf{G})}{p(\mathcal{D})} d\mathbf{Z} \tag{A.3}$$

$$= \sum_{\mathbf{G}} \int_{\mathbf{Z}} \frac{p(\mathbf{Z})p(\mathbf{G} \mid \mathbf{Z})p(\mathcal{D} \mid \mathbf{G}) f(\mathbf{G})}{p(\mathcal{D})} d\mathbf{Z} \tag{A.4}$$

by the generative model in (5)

$$= \sum_{\mathbf{G}} \int_{\mathbf{Z}} \frac{p(\mathbf{Z} \mid \mathcal{D})p(\mathbf{G} \mid \mathbf{Z})p(\mathcal{D} \mid \mathbf{G}) f(\mathbf{G})}{p(\mathcal{D} \mid \mathbf{Z})} d\mathbf{Z} \tag{A.5}$$

since $p(\mathbf{Z} \mid \mathcal{D}) = \dfrac{p(\mathbf{Z})p(\mathcal{D} \mid \mathbf{Z})}{p(\mathcal{D})}$

$$= \int_{\mathbf{Z}} p(\mathbf{Z} \mid \mathcal{D}) \frac{\sum_{\mathbf{G}} p(\mathbf{G} \mid \mathbf{Z})p(\mathcal{D} \mid \mathbf{G}) f(\mathbf{G})}{p(\mathcal{D} \mid \mathbf{Z})} d\mathbf{Z} \qquad \text{rearranging} \tag{A.6}$$

$$= \int_{\mathbf{Z}} p(\mathbf{Z} \mid \mathcal{D}) \frac{\sum_{\mathbf{G}} p(\mathbf{G} \mid \mathbf{Z})p(\mathcal{D} \mid \mathbf{G}) f(\mathbf{G})}{\sum_{\mathbf{G}} p(\mathbf{G}, \mathcal{D} \mid \mathbf{Z})} d\mathbf{Z} \tag{A.7}$$

by the law of total probability

$$= \int_{\mathbf{Z}} p(\mathbf{Z} \mid \mathcal{D}) \frac{\sum_{\mathbf{G}} p(\mathbf{G} \mid \mathbf{Z})p(\mathcal{D} \mid \mathbf{G}) f(\mathbf{G})}{\sum_{\mathbf{G}} p(\mathbf{G} \mid \mathbf{Z})p(\mathcal{D} \mid \mathbf{G})} d\mathbf{Z} \tag{A.8}$$

expanding $p(\mathbf{G}, \mathcal{D} \mid \mathbf{Z})$ by the generative model in (5)

$$= \mathbb{E}_{p(\mathbf{Z} \mid \mathcal{D})} \left[ \frac{\mathbb{E}_{p(\mathbf{G} \mid \mathbf{Z})} \big[ f(\mathbf{G})p(\mathcal{D} \mid \mathbf{G}) \big]}{\mathbb{E}_{p(\mathbf{G} \mid \mathbf{Z})} \big[ p(\mathcal{D} \mid \mathbf{G}) \big]} \right] \tag{A.9}$$

as desired for (a).

Finally, let us consider (b), the general case. The derivation essentially follows the same ideas as for (a) but does not marginalize out $\mathbf{\Theta}$.

$$\mathbb{E}_{p(\mathbf{G}, \mathbf{\Theta} \mid D)} f(\mathbf{G}, \mathbf{\Theta}) \tag{A.10}$$

$$= \sum_{\mathbf{G}} \int_{\mathbf{\Theta}} p(\mathbf{G}, \mathbf{\Theta} \mid \mathcal{D}) f(\mathbf{G}, \mathbf{\Theta}) d\mathbf{\Theta} \tag{A.11}$$

$$= \sum_{\mathbf{G}} \int_{\mathbf{\Theta}} \frac{p(\mathbf{G}, \mathbf{\Theta}, \mathcal{D}) f(\mathbf{G}, \mathbf{\Theta})}{p(\mathcal{D})} d\mathbf{\Theta} \tag{A.12}$$

$$= \sum_{\mathbf{G}} \int_{\mathbf{\Theta}} \int_{\mathbf{Z}} \frac{p(\mathbf{Z}, \mathbf{G}, \mathbf{\Theta}, \mathcal{D}) f(\mathbf{G}, \mathbf{\Theta})}{p(\mathcal{D})} d\mathbf{Z} d\mathbf{\Theta} \tag{A.13}$$

$$= \sum_{\mathbf{G}} \int_{\Theta} \int_{\mathbf{Z}} \frac{p(\mathbf{Z})p(\mathbf{G}\,|\,\mathbf{Z})p(\Theta\,|\,\mathbf{G})p(\mathcal{D}\,|\,\mathbf{G},\Theta)f(\mathbf{G},\Theta)}{p(\mathcal{D})} d\mathbf{Z}d\Theta \tag{A.14}$$

by the generative model in (5)

$$= \sum_{\mathbf{G}} \int_{\Theta} \int_{\mathbf{Z}} \frac{p(\mathbf{Z},\Theta\,|\,\mathcal{D})p(\mathbf{G}\,|\,\mathbf{Z})p(\Theta\,|\,\mathbf{G})p(\mathcal{D}\,|\,\mathbf{G},\Theta)f(\mathbf{G},\Theta)}{p(\mathcal{D},\Theta\,|\,\mathbf{Z})} d\mathbf{Z}d\Theta \tag{A.15}$$

since $p(\mathbf{Z},\Theta\,|\,\mathcal{D}) = \dfrac{p(\mathbf{Z})p(\Theta,\mathcal{D}\,|\,\mathbf{Z})}{p(\mathcal{D})}$

$$= \int_{\Theta} \int_{\mathbf{Z}} p(\mathbf{Z},\Theta\,|\,\mathcal{D}) \frac{\sum_{\mathbf{G}} p(\mathbf{G}\,|\,\mathbf{Z})p(\Theta\,|\,\mathbf{G})p(\mathcal{D}\,|\,\mathbf{G},\Theta)f(\mathbf{G},\Theta)}{p(\Theta,\mathcal{D}\,|\,\mathbf{Z})} d\mathbf{Z}d\Theta \tag{A.16}$$

rearranging

$$= \int_{\Theta} \int_{\mathbf{Z}} p(\mathbf{Z},\Theta\,|\,\mathcal{D}) \frac{\sum_{\mathbf{G}} p(\mathbf{G}\,|\,\mathbf{Z})p(\Theta\,|\,\mathbf{G})p(\mathcal{D}\,|\,\mathbf{G},\Theta)f(\mathbf{G},\Theta)}{\sum_{\mathbf{G}} p(\mathbf{G},\Theta,\mathcal{D}\,|\,\mathbf{Z})} d\mathbf{Z}d\Theta \tag{A.17}$$

by the law of total probability

$$= \int_{\Theta} \int_{\mathbf{Z}} p(\mathbf{Z},\Theta\,|\,\mathcal{D}) \frac{\sum_{\mathbf{G}} p(\mathbf{G}\,|\,\mathbf{Z})p(\Theta\,|\,\mathbf{G})p(\mathcal{D}\,|\,\mathbf{G},\Theta)f(\mathbf{G},\Theta)}{\sum_{\mathbf{G}} p(\mathbf{G}\,|\,\mathbf{Z})p(\Theta\,|\,\mathbf{G})p(\mathcal{D}\,|\,\mathbf{G},\Theta)} d\mathbf{Z}d\Theta \tag{A.18}$$

expanding $p(\mathbf{G},\Theta,\mathcal{D}\,|\,\mathbf{Z})$ by the generative model in (5)

$$= \mathbb{E}_{p(\mathbf{Z},\Theta\,|\,\mathcal{D})} \left[ \frac{\mathbb{E}_{p(\mathbf{G}\,|\,\mathbf{Z})} \big[ f(\mathbf{G},\Theta)p(\Theta\,|\,\mathbf{G})p(\mathcal{D}\,|\,\mathbf{G},\Theta) \big]}{\mathbb{E}_{p(\mathbf{G}\,|\,\mathbf{Z})} \big[ p(\Theta\,|\,\mathbf{G})p(\mathcal{D}\,|\,\mathbf{G},\Theta) \big]} \right] \tag{A.19}$$

which is the statement in (b).  □

## A.2  Proposition 2

**Proof**   We will derive the gradients of the unnormalized posterior since

$$\nabla_{\mathbf{Z}} \log p(\mathbf{Z}\,|\,\mathcal{D}) = \nabla_{\mathbf{Z}} \log p(\mathbf{Z},\mathcal{D}) - \nabla_{\mathbf{Z}} \log p(\mathcal{D}) = \nabla_{\mathbf{Z}} \log p(\mathbf{Z},\mathcal{D}) \tag{A.20}$$

and analogously for the other two expressions. Through straightforward manipulation and using the identity $\nabla_{\mathbf{x}} \log f(\mathbf{x}) = \nabla_{\mathbf{x}} f(\mathbf{x})/f(\mathbf{x})$, we obtain

$$\nabla_{\mathbf{Z}} \log p(\mathbf{Z},\mathcal{D}) = \nabla_{\mathbf{Z}} \log p(\mathbf{Z}) + \nabla_{\mathbf{Z}} \log p(\mathcal{D}\,|\,\mathbf{Z}) \tag{A.21}$$

$$= \nabla_{\mathbf{Z}} \log p(\mathbf{Z}) + \frac{\nabla_{\mathbf{Z}}\, p(\mathcal{D}\,|\,\mathbf{Z})}{p(\mathcal{D}\,|\,\mathbf{Z})} \tag{A.22}$$

$$= \nabla_{\mathbf{Z}} \log p(\mathbf{Z}) + \frac{\nabla_{\mathbf{Z}}\big[\sum_{\mathbf{G}} p(\mathbf{G}\,|\,\mathbf{Z})p(\mathcal{D}\,|\,\mathbf{G})\big]}{\sum_{\mathbf{G}} p(\mathbf{G}\,|\,\mathbf{Z})p(\mathcal{D}\,|\,\mathbf{G})} \tag{A.23}$$

$$= \nabla_{\mathbf{Z}} \log p(\mathbf{Z}) + \frac{\nabla_{\mathbf{Z}}\, \mathbb{E}_{p(\mathbf{G}\,|\,\mathbf{Z})}[p(\mathcal{D}\,|\,\mathbf{G})]}{\mathbb{E}_{p(\mathbf{G}\,|\,\mathbf{Z})}[p(\mathcal{D}\,|\,\mathbf{G})]} \tag{A.24}$$

Analogously, we get

$$\nabla_{\mathbf{Z}} \log p(\mathbf{Z},\Theta,\mathcal{D}) = \nabla_{\mathbf{Z}} \log p(\mathbf{Z}) + \nabla_{\mathbf{Z}} \log p(\Theta,\mathcal{D}\,|\,\mathbf{Z}) \tag{A.25}$$

$$= \nabla_{\mathbf{Z}} \log p(\mathbf{Z}) + \frac{\nabla_{\mathbf{Z}}\, p(\Theta,\mathcal{D}\,|\,\mathbf{Z})}{p(\Theta,\mathcal{D}\,|\,\mathbf{Z})} \tag{A.26}$$

$$= \nabla_{\mathbf{Z}} \log p(\mathbf{Z}) + \frac{\nabla_{\mathbf{Z}}\big[\sum_{\mathbf{G}} p(\mathbf{G}\,|\,\mathbf{Z})p(\Theta,\mathcal{D}\,|\,\mathbf{G})\big]}{\sum_{\mathbf{G}} p(\mathbf{G}\,|\,\mathbf{Z})p(\Theta,\mathcal{D}\,|\,\mathbf{G})} \tag{A.27}$$

$$= \nabla_{\mathbf{Z}} \log p(\mathbf{Z}) + \frac{\nabla_{\mathbf{Z}} \, \mathbb{E}_{p(\mathbf{G}\,|\,\mathbf{Z})}[p(\mathbf{\Theta}, \mathcal{D}\,|\,\mathbf{G})]}{\mathbb{E}_{p(\mathbf{G}\,|\,\mathbf{Z})}[p(\mathbf{\Theta}, \mathcal{D}\,|\,\mathbf{G})]} \tag{A.28}$$

Lastly, using the same ideas as above, we arrive at

$$\nabla_{\mathbf{\Theta}} \log p(\mathbf{Z}, \mathbf{\Theta}, \mathcal{D}) = \nabla_{\mathbf{\Theta}} \log p(\mathbf{Z}) + \nabla_{\mathbf{\Theta}} \log p(\mathbf{\Theta}, \mathcal{D}\,|\,\mathbf{Z}) \tag{A.29}$$

$$= \frac{\nabla_{\mathbf{\Theta}} \, p(\mathbf{\Theta}, \mathcal{D}\,|\,\mathbf{Z})}{p(\mathbf{\Theta}, \mathcal{D}\,|\,\mathbf{Z})} \tag{A.30}$$

$$= \frac{\nabla_{\mathbf{\Theta}} \big[ \sum_{\mathbf{G}} p(\mathbf{G}\,|\,\mathbf{Z}) p(\mathbf{\Theta}, \mathcal{D}\,|\,\mathbf{G}) \big]}{\sum_{\mathbf{G}} p(\mathbf{G}\,|\,\mathbf{Z}) p(\mathbf{\Theta}, \mathcal{D}\,|\,\mathbf{G})} \tag{A.31}$$

$$= \frac{\sum_{\mathbf{G}} p(\mathbf{G}\,|\,\mathbf{Z}) \nabla_{\mathbf{\Theta}} p(\mathbf{\Theta}, \mathcal{D}\,|\,\mathbf{G})}{\sum_{\mathbf{G}} p(\mathbf{G}\,|\,\mathbf{Z}) p(\mathbf{\Theta}, \mathcal{D}\,|\,\mathbf{G})} \tag{A.32}$$

$$= \frac{\mathbb{E}_{p(\mathbf{G}\,|\,\mathbf{Z})} \big[ \nabla_{\mathbf{\Theta}} p(\mathbf{\Theta}, \mathcal{D}\,|\,\mathbf{G}) \big]}{\mathbb{E}_{p(\mathbf{G}\,|\,\mathbf{Z})}[p(\mathbf{\Theta}, \mathcal{D}\,|\,\mathbf{G})]} \quad \square \tag{A.33}$$

In the above, without any additional factor modeling a prior belief over graphs, the score of the latent prior $p_\beta(\mathbf{Z})$ as defined in (8) is given by

$$\nabla_{\mathbf{Z}} \log p_\beta(\mathbf{Z}) = -\beta \, \nabla_{\mathbf{Z}} \, \mathbb{E}_{p(\mathbf{G}\,|\,\mathbf{Z})}[h(\mathbf{G})] - \frac{1}{\sigma_z^2} \mathbf{Z} \tag{A.34}$$

**Practical considerations**  Estimating expectations of the form $\mathbb{E}_{p(\mathbf{G}\,|\,\mathbf{Z})}[f(\mathbf{G})]$ with Monte Carlo sampling can be numerically challenging when $f$ are probability densities and thus often close to zero. In practice, we recommend the log-sum-exp trick for applying Proposition 2. Let us define

$$\underset{m=1}{\overset{M}{\text{L}\Sigma\text{E}}}\{x^{(m)}\} := \log \left( \sum_{m=1}^{M} \exp\left(x^{(m)}\right) \right) \tag{A.35}$$

For $M$ Monte Carlo samples $\mathbf{G}^{(m)} \sim p(\mathbf{G}\,|\,\mathbf{Z})$, we can rewrite the estimator for the expectation as

$$\mathbb{E}_{p(\mathbf{G}\,|\,\mathbf{Z})}[f(\mathbf{G})] \approx \frac{1}{M} \sum_{m=1}^{M} f(\mathbf{G}^{(m)}) = \exp\left( \log\left( \sum_{m=1}^{M} f(\mathbf{G}^{(m)}) \right) - \log M \right) \tag{A.36}$$

$$= \exp\left( \underset{m=1}{\overset{M}{\text{L}\Sigma\text{E}}} \left\{ \log f(\mathbf{G}^{(m)}) \right\} - \log M \right) \tag{A.37}$$

Computing L$\Sigma$E can be made numerically stable by subtracting and adding $\max_m\{x^{(m)}\}$ before and after applying L$\Sigma$E to $\{x^{(m)}\}$, respectively. Stable L$\Sigma$E can be extended to handle negative-valued $f$ inside the expectation, e.g., for the gradient of $f$, and to the ratios of expectations in Proposition 2.

### A.3   Equations (16) and (17)

**Proof**  The sigmoid function converges to the unit step function, i.e. $\sigma_\alpha(x) \to \mathbb{1}[x > 0]$ as $\alpha \to \infty$. Hence, the edge probabilities $\mathbf{G}_\alpha(\mathbf{Z})$ defined in (7) converge to a (binary) matrix $\mathbf{G}$ as $\alpha \to \infty$. Extending the notation of (7), we will denote this single limiting graph implied by $\mathbf{Z}$ as $\mathbf{G}_\infty(\mathbf{Z})$ where

$$\mathbf{G}_\infty(\mathbf{Z})_{ij} := \begin{cases} 1 & \text{if } \mathbf{u}_i^\top \mathbf{v}_j > 0 \text{ and } i \neq j \\ 0 & \text{otherwise} \end{cases}$$

The above implies that when the temperature parameter $\alpha$ is annealed to $\infty$, the probability mass function and correspondingly the expectation simplify:

$$\text{As } \alpha \to \infty : \quad p_\alpha(\mathbf{G}\,|\,\mathbf{Z}) \to \mathbb{1}[\mathbf{G} = \mathbf{G}_\infty(\mathbf{Z})] \tag{A.38}$$
$$\mathbb{E}_{p_\alpha(\mathbf{G}\,|\,\mathbf{Z})}[f(\mathbf{G})] \to f(\mathbf{G}_\infty(\mathbf{Z}))$$

Again, let us first consider case (a). Starting with Proposition 1(a) in the first step, we can use the above insight to simplify the inner expectations:

$$\mathbb{E}_{p(\mathbf{G}\,|\,\mathcal{D})}\Big[f(\mathbf{G})\Big] = \mathbb{E}_{p(\mathbf{Z}\,|\,\mathcal{D})} \left[ \frac{\mathbb{E}_{p_\alpha(\mathbf{G}\,|\,\mathbf{Z})}\big[f(\mathbf{G}) p(\mathcal{D}\,|\,\mathbf{G})\big]}{\mathbb{E}_{p_\alpha(\mathbf{G}\,|\,\mathbf{Z})}\big[p(\mathcal{D}\,|\,\mathbf{G})\big]} \right] \tag{A.39}$$

$$\xrightarrow{\alpha \to \infty} \mathbb{E}_{p(\mathbf{Z} \mid \mathcal{D})} \left[ \frac{f(\mathbf{G}_\infty(\mathbf{Z}))p(\mathcal{D} \mid \mathbf{G}_\infty(\mathbf{Z}))}{p(\mathcal{D} \mid \mathbf{G}_\infty(\mathbf{Z}))} \right] \tag{A.40}$$

$$= \mathbb{E}_{p(\mathbf{Z} \mid \mathcal{D})} \left[ f(\mathbf{G}_\infty(\mathbf{Z})) \right] \tag{A.41}$$

Analogously, we get for the general case (b):

$$\mathbb{E}_{p(\mathbf{G}, \mathbf{\Theta} \mid \mathcal{D})} \left[ f(\mathbf{G}, \mathbf{\Theta}) \right] = \mathbb{E}_{p(\mathbf{Z}, \mathbf{\Theta} \mid \mathcal{D})} \left[ \frac{\mathbb{E}_{p_\alpha(\mathbf{G} \mid \mathbf{Z})} \left[ f(\mathbf{G}, \mathbf{\Theta})p(\mathbf{\Theta} \mid \mathbf{G})p(\mathcal{D} \mid \mathbf{G}, \mathbf{\Theta}) \right]}{\mathbb{E}_{p_\alpha(\mathbf{G} \mid \mathbf{Z})} \left[ p(\mathbf{\Theta} \mid \mathbf{G})p(\mathcal{D} \mid \mathbf{G}, \mathbf{\Theta}) \right]} \right] \tag{A.42}$$

$$\xrightarrow{\alpha \to \infty} \mathbb{E}_{p(\mathbf{Z}, \mathbf{\Theta} \mid \mathcal{D})} \left[ \frac{f(\mathbf{G}_\infty(\mathbf{Z}), \mathbf{\Theta})p(\mathbf{\Theta} \mid \mathbf{G}_\infty(\mathbf{Z}))p(\mathcal{D} \mid \mathbf{G}_\infty(\mathbf{Z}), \mathbf{\Theta})}{p(\mathbf{\Theta} \mid \mathbf{G}_\infty(\mathbf{Z}))p(\mathcal{D} \mid \mathbf{G}_\infty(\mathbf{Z}), \mathbf{\Theta})} \right] \tag{A.43}$$

$$= \mathbb{E}_{p(\mathbf{Z}, \mathbf{\Theta} \mid \mathcal{D})} \left[ f(\mathbf{G}_\infty(\mathbf{Z}), \mathbf{\Theta}) \right] \quad \square \tag{A.44}$$

# B  Gradient Estimation for Bayesian Inference

To derive the expressions for the gradient estimators in (12) and (14) for both the marginal likelihood and the likelihood cases, we will use a generic function $f(\mathbf{G})$ as a placeholder for either $p(\mathcal{D} \mid \mathbf{G})$ or $p(\mathcal{D} \mid \mathbf{G}, \mathbf{\Theta})$, since the results hold for general densities $f(\mathbf{G})$.

## B.1  Gumbel-Softmax Estimator for the Likelihood Gradient

In general, for a Bernoulli random variable $X$ with $p(X = 1) = q$, it holds that

$$X \stackrel{d}{=} \mathbb{1} \left[ G_1 + \log q > G_0 + \log(1 - q) \right] \tag{B.1}$$

when $G_0, G_1 \sim \text{Gumbel}(0, 1)$. This is the Gumbel-*max* trick. Since the unit step function $\mathbb{1}[\cdot]$ does not have an informative gradient, Maddison et al. [48] and Jang et al. [49] have proposed to use the sigmoid function $\sigma_\tau(\cdot)$ with parameter $\tau$ as a soft relaxation of $\mathbb{1}[\cdot]$.

Using this so-called Gumbel-*softmax* trick, we can reparameterize the entries of $\mathbf{G}$ under the graph model in (6). Starting from the Gumbel-max equality in (B.1), we rearrange the inequality inside the indicator into the form "$> 0$" and apply the sigmoid relaxation with parameter $\tau$. We obtain the following soft relaxation for each entry of $\mathbf{G}$:

$$g_{ij} \approx \sigma_\tau \left( G_1 - G_0 + \log \sigma_\alpha(\mathbf{u}_i^\top \mathbf{v}_j) - \log(1 - \sigma_\alpha(\mathbf{u}_i^\top \mathbf{v}_j)) \right) \tag{B.2}$$

$$= \sigma_\tau \left( L + \log \left( \frac{\sigma_\alpha(\mathbf{u}_i^\top \mathbf{v}_j)}{\sigma_\alpha(-\mathbf{u}_i^\top \mathbf{v}_j)} \right) \right) \tag{B.3}$$

$$= \sigma_\tau \left( L + \log \left( \frac{\exp(\alpha \mathbf{u}_i^\top \mathbf{v}_j)}{\exp(\alpha \mathbf{u}_i^\top \mathbf{v}_j) + 1} \frac{\exp(\alpha \mathbf{u}_i^\top \mathbf{v}_j) + 1}{1} \right) \right) \tag{B.4}$$

$$= \sigma_\tau \left( L + \log \left( \exp(\alpha \mathbf{u}_i^\top \mathbf{v}_j) \right) \right) \tag{B.5}$$

$$= \sigma_\tau \left( L + \alpha \mathbf{u}_i^\top \mathbf{v}_j \right) \tag{B.6}$$

where $L \sim \text{Logistic}(0, 1)$ since $L \stackrel{d}{=} G_1 - G_0$ when $G_0, G_1 \sim \text{Gumbel}(0, 1)$. For $i = j$, we set $g_{ij} := 0$ by default in accordance with the graph model in (6). Since this allows us to separate the randomness in sampling from the distribution $p(\mathbf{G} \mid \mathbf{Z})$ from the values of $\mathbf{Z}$, we can move the gradient operator inside the expectation and obtain the estimator given in (12):

$$\nabla_{\mathbf{Z}} \mathbb{E}_{p(\mathbf{G} \mid \mathbf{Z})} \left[ f(\mathbf{G}) \right] \approx \mathbb{E}_{p(\mathbf{L})} \left[ \nabla_{\mathbf{Z}} f(\mathbf{G}_\tau(\mathbf{L}, \mathbf{Z})) \right]$$

$$= \mathbb{E}_{p(\mathbf{L})} \left[ \nabla_{\mathbf{G}} f(\mathbf{G}) \big|_{\mathbf{G} = \mathbf{G}_\tau(\mathbf{L}, \mathbf{Z})} \cdot \nabla_{\mathbf{Z}} \mathbf{G}_\tau(\mathbf{L}, \mathbf{Z}) \right] \tag{B.7}$$

While the reparameterization trick generally provides a lower variance estimate of the gradient, the form in (12) is biased when $\tau < \infty$ because we use a soft relaxation of the true distribution. In addition, the estimator in (12) requires that $\nabla_{\mathbf{G}} p(\mathcal{D} \mid \mathbf{G})$ or $\nabla_{\mathbf{G}} p(\mathbf{\Theta}, \mathcal{D} \mid \mathbf{G})$ is available, depending

on the inference task. In case $p(\mathcal{D} \,|\, \mathbf{G})$ or $p(\mathbf{\Theta}, \mathcal{D} \,|\, \mathbf{G})$ is only defined for discrete $\mathbf{G}$, it is possible to evaluate $\nabla_{\mathbf{G}} p(\mathcal{D} \,|\, \mathbf{G})$ or $\nabla_{\mathbf{G}} p(\mathbf{\Theta}, \mathcal{D} \,|\, \mathbf{G})$ using hard Gumbel-max samples of $\mathbf{G}$ (i.e., with $\tau = \infty$). As before, however, one would use soft Gumbel-softmax samples in $\nabla_{\mathbf{Z}} \mathbf{G}_{\tau}(\mathbf{L}, \mathbf{Z})$ to obtain an informative gradient. Lastly, we can use this estimator to approximate the score of the latent prior $\nabla_{\mathbf{Z}} \log p(\mathbf{Z})$ given in (8) because the acyclicity constraint $h(\mathbf{G})$ is differentiable with respect to $\mathbf{G}$. In practice, the log-sum-exp trick described in Section A.2 as well as the score function identity $\nabla_{\mathbf{x}} f(\mathbf{x}) = f(\mathbf{x}) \nabla_{\mathbf{x}} \log f(\mathbf{x})$ should be used for numerically stable computation of the estimator.

### B.2 Score Function Estimator for the Likelihood Gradient

To arrive at the estimator in (14), we expand the gradient expression as

$$\nabla_{\mathbf{Z}} \mathbb{E}_{p(\mathbf{G} \,|\, \mathbf{Z})}\big[f(\mathbf{G})\big] = \sum_{\mathbf{G}} f(\mathbf{G}) \nabla_{\mathbf{Z}} p(\mathbf{G} \,|\, \mathbf{Z}) = \sum_{\mathbf{G}} f(\mathbf{G}) p(\mathbf{G} \,|\, \mathbf{Z}) \nabla_{\mathbf{Z}} \log p(\mathbf{G} \,|\, \mathbf{Z}) \tag{B.8}$$

$$= \mathbb{E}_{p(\mathbf{G} \,|\, \mathbf{Z})}\Big[f(\mathbf{G}) \, \nabla_{\mathbf{Z}} \log p(\mathbf{G} \,|\, \mathbf{Z})\Big] \tag{B.9}$$

where (B.8) uses the fact that $\nabla_{\mathbf{Z}} \log p(\mathbf{G} \,|\, \mathbf{Z}) = \nabla_{\mathbf{Z}} p(\mathbf{G} \,|\, \mathbf{Z}) / p(\mathbf{G} \,|\, \mathbf{Z})$. Finally, we recall the well-known property of the score function that $\mathbb{E}_{p(\mathbf{G} \,|\, \mathbf{Z})}[\nabla_{\mathbf{Z}} \log p(\mathbf{G} \,|\, \mathbf{Z})] = \mathbf{0}$. Due to this, for any constant $b$ as written in (14), the estimator is unbiased because the additional term involving $b$ has zero expectation. The constant can be used to reduce the variance of the Monte Carlo estimator [51]. In our experiments, we always use $b = 0$.

## C  Background: Stein Variational Gradient Descent

This section describes Stein variational gradient descent (SVGD) by Liu and Wang [19]. The overview is meant as supplementary material for Section 5, where we propose to use SVGD for inferring the DiBS posteriors $p(\mathbf{Z} \,|\, \mathcal{D})$ and $p(\mathbf{Z}, \mathbf{\Theta} \,|\, \mathcal{D})$. In contrast to sampling-based MCMC or optimization-based variational inference methods, SVGD iteratively *transports* a fixed set of particles to closely match a target distribution, akin to the gradient descent algorithm in optimization. We refer the reader to Liu and Wang [19] for additional details.

Let $p(\mathbf{x})$ with $\mathbf{x} \in \mathcal{X}$ be a differentiable density that we want to sample from, e.g., to estimate an expectation. Starting from a smooth reference density $q(\mathbf{x})$, SVGD aims to find a one-to-one transform $\mathbf{t} : \mathcal{X} \mapsto \mathcal{X}$ such that the transformed density $q_{[\mathbf{t}]}(\widetilde{\mathbf{x}})$ with $\widetilde{\mathbf{x}} := \mathbf{t}(\mathbf{x})$ minimizes the KL-divergence to $p$. In particular, Liu and Wang [19] propose to use the incremental transform

$$\mathbf{t}(\mathbf{x}) = \mathbf{x} + \eta \, \phi(\mathbf{x}) \tag{C.1}$$

When $|\eta|$ is sufficiently small, the Jacobian of $\mathbf{t}$ has full rank and $\mathbf{t}$ is one-to-one. The key result by Liu and Wang [19] links the incremental transform $\mathbf{t}$ in (C.1) to prior work on reproducing kernel Hilbert spaces (RKHSs). The authors show that if $\phi$ lies in the unit ball of the RKHS induced by a kernel $k$, then the transform $\mathbf{t}$ maximizing the descent on the KL divergence from $q_{[\mathbf{t}]}$ to $p$ uses an incremental update $\phi$ proportional to

$$\phi_{q,p}^{*}(\cdot) = \mathbb{E}_{q(\mathbf{x})} \big[ k(\mathbf{x}, \cdot) \nabla_{\mathbf{x}} \log p(\mathbf{x})^{\top} + \nabla_{\mathbf{x}} k(\mathbf{x}, \cdot) \big] \tag{C.2}$$

This suggests an iterative procedure of repeatedly applying the update of (C.1) with $\phi = \phi_{q,p}^{*}(\cdot)$ from (C.2) to a finite set of randomly initialized particles $\{\mathbf{x}^{(m)}\}_{m=1}^{M}$. At each iteration $t$, the $m$-th particle $\mathbf{x}^{(m)}$ is then deterministically updated according to:

$$\mathbf{x}_{t+1}^{(m)} \leftarrow \mathbf{x}_{t}^{(m)} + \eta_t \, \phi(\mathbf{x}_t^{(m)})$$
$$\text{where } \phi(\mathbf{x}) = \frac{1}{M} \sum_{k=1}^{M} \Big[ k(\mathbf{x}_t^{(k)}, \mathbf{x}) \, \nabla_{\mathbf{x}_t^{(k)}} \log p(\mathbf{x}_t^{(k)}) + \nabla_{\mathbf{x}_t^{(k)}} k(\mathbf{x}_t^{(k)}, \mathbf{x}) \Big] \tag{C.3}$$

For sufficiently small step sizes $\eta_t$, the sequence of particles eventually converges, in which case the transform $\mathbf{t}$ reduces to the identity mapping. The particle update in (C.3) consists of a gradient ascent term driving the particles to high-density regions, and a term involving $\nabla_{\mathbf{x}} k(\mathbf{x}, \cdot)$ that acts as a repulsive force between particles, preventing them from collapsing into the modes of $p(\mathbf{x})$.

## D General Algorithm

---

**Algorithm 2** DiBS with SVGD [19] for inference of $p(\mathbf{G}, \boldsymbol{\Theta} \,|\, \mathcal{D})$

---

    **Input:** Initial latent and parameter particles $\{(\mathbf{Z}_0^{(m)}, \boldsymbol{\Theta}_0^{(m)})\}_{m=1}^M$, kernel $k$, schedules for $\eta_t, \alpha_t, \beta_t$
    **Output:** Set of graph and parameter particles $\{(\mathbf{G}^{(m)}, \boldsymbol{\Theta}^{(m)})\}_{m=1}^M$ approximating $p(\mathbf{G}, \boldsymbol{\Theta} \,|\, \mathcal{D})$

1: Incorporate prior belief of $p(\mathbf{G})$ into $p(\mathbf{Z})$         ▷ See Section 4.2
2: **for** iteration $t = 0$ to $T - 1$ **do**
3:     Estimate score $\nabla_{\mathbf{Z}} \log p(\mathbf{Z}, \boldsymbol{\Theta} \,|\, \mathcal{D})$ given in (10) for each $\mathbf{Z}_t^{(m)}$     ▷ See (12) and (14)
4:     Estimate score $\nabla_{\boldsymbol{\Theta}} \log p(\mathbf{Z}, \boldsymbol{\Theta} \,|\, \mathcal{D})$ given in (11) for each $\boldsymbol{\Theta}_t^{(m)}$
5:     **for** particle $m = 1$ to $M$ **do**
6:         $\mathbf{Z}_{t+1}^{(m)} \leftarrow \mathbf{Z}_t^{(m)} + \eta_t \, \boldsymbol{\phi}_t^{\mathbf{Z}}(\mathbf{Z}_t^{(m)}, \boldsymbol{\Theta}_t^{(m)})$     ▷ SVGD step

$$\text{where } \boldsymbol{\phi}_t^{\mathbf{Z}}(\cdot, \cdot) := \frac{1}{M} \sum_{k=1}^M \Big[ k\big((\mathbf{Z}_t^{(k)}, \boldsymbol{\Theta}_t^{(k)}), (\cdot, \cdot)\big) \, \nabla_{\mathbf{z}_t^{(k)}} \log p\big(\mathbf{Z}_t^{(k)}, \boldsymbol{\Theta}_t^{(k)} \,|\, \mathcal{D}\big)$$
$$+ \nabla_{\mathbf{z}_t^{(k)}} k\big((\mathbf{Z}_t^{(k)}, \boldsymbol{\Theta}_t^{(k)}), (\cdot, \cdot)\big) \Big]$$

7:         $\boldsymbol{\Theta}_{t+1}^{(m)} \leftarrow \boldsymbol{\Theta}_t^{(m)} + \eta_t \, \boldsymbol{\phi}_t^{\boldsymbol{\Theta}}(\mathbf{Z}_t^{(m)}, \boldsymbol{\Theta}_t^{(m)})$
        where $\boldsymbol{\phi}_t^{\boldsymbol{\Theta}}(\cdot, \cdot)$ is analogous to $\boldsymbol{\phi}_t^{\mathbf{Z}}(\cdot, \cdot)$ but using gradients $\nabla_{\boldsymbol{\Theta}_t^{(k)}}$ instead of $\nabla_{\mathbf{z}_t^{(k)}}$
8: **return** $\{(\mathbf{G}_\infty(\mathbf{Z}_T^{(m)}), \boldsymbol{\Theta}_T^{(m)})\}_{m=1}^M$     ▷ See (16) and (17)

---

## E Experimental Details

### E.1 Gaussian Bayesian Networks

In our experiments, we consider Bayesian networks with Gaussian local conditional distributions of each variable given its parents. For both linear or nonlinear Gaussian BNs, which will be defined presently, the generative model for synthetic data simulation as well as the parameter prior used for joint inference are set to standard Gaussian distributions. We fix the observation noise to $\sigma^2 = 0.1$ for all nodes both during synthetic data generation and joint posterior inference, rendering the causal structure fully identifiable [61].

**Linear** Analogous to linear regression, linear Gaussian BNs model the mean of a given variable as a linear function of its parents:

$$p(\mathbf{x} \,|\, \mathbf{G}, \boldsymbol{\Theta}) = \prod_{i=1}^d \mathcal{N}(x_i; \boldsymbol{\theta}_i^\top \mathbf{x}_{\text{pa}(i)}, \sigma^2) \tag{E.1}$$
$$\text{or} \quad p(\mathbf{x} \,|\, \mathbf{G}, \boldsymbol{\Theta}) = \mathcal{N}\big(\mathbf{x}; (\mathbf{G} \circ \boldsymbol{\Theta})^\top \mathbf{x}, \sigma^2 \mathbf{I}\big)$$

where "$\circ$" denotes elementwise multiplication. In our experiments, DiBS uses the second parameterization in (E.1) to allow for a constant dimensionality of the conditional distribution parameters $\boldsymbol{\Theta}$ and make the likelihood well-defined for the Gumbel-softmax estimator in (12).

When inferring the marginal posterior $p(\mathbf{G} \,|\, \mathcal{D})$ for linear Gaussian BNs, we follow the predominant choice in the literature and employ the *Bayesian Gaussian Equivalent (BGe)* marginal likelihood, under which Markov equivalent structures are scored equally [16, 17]. Details on the computation of the BGe score are provided by Kuipers et al. [62]. Following the notation of Geiger and Heckerman [17] and Kuipers et al. [62], we use the standard effective sample size hyperparameters $\alpha_\mu = 1$ and $\alpha_\omega = d + 2$ as well as the diagonal form of the Wishart inverse scale matrix for the Normal-Wishart parameter prior underlying the BGe score.

**Nonlinear** The interaction between variables $\mathbf{x}$ can straightforwardly be extended to be *nonlinear*, e.g., using neural networks. In Section 6.3, we follow Zheng et al. [27] and consider (fully connected) feed-forward neural networks (FFNs) of the form

$$\text{FFN}(\cdot\,; \boldsymbol{\Theta}) : \mathbb{R}^d \to \mathbb{R}$$
$$\text{FFN}(\mathbf{u}; \boldsymbol{\Theta}) := \boldsymbol{\Theta}^{(L)} \sigma\Big( \dots \boldsymbol{\Theta}^{(2)} \sigma\big(\boldsymbol{\Theta}^{(1)} \mathbf{u} + \boldsymbol{\theta}_b^{(1)}\big) + \boldsymbol{\theta}_b^{(2)} \dots \Big) + \boldsymbol{\theta}_b^{(L)} \tag{E.2}$$

with weights $\mathbf{\Theta}^{(l)} \in \mathbb{R}^{d_l \times d_{l-1}}$, biases $\boldsymbol{\theta}_b^{(l)} \in \mathbb{R}^{d_l}$, and elementwise activation function $\sigma : \mathbb{R} \to \mathbb{R}$. Zheng et al. [27] show that the class of fully connected neural networks in (E.2) that do *not* depend on the value of $u_k$ is equivalent to the class of fully connected neural networks in (E.2) where the $k$-th column of $\mathbf{\Theta}^{(1)}$ equals zero. This insight allows us to define a nonlinear Gaussian BN parameterized by a fully connected neural network:

$$p(\mathbf{x} \,|\, \mathbf{G}, \mathbf{\Theta}) = \prod_{i=1}^{d} \mathcal{N}\Big(x_i;\ \mathrm{FFN}\big(\mathbf{G}_i^{\top} \circ \mathbf{x}; \mathbf{\Theta}_i\big),\ \sigma^2\Big) \tag{E.3}$$

As required for a BN, each variable is independent of its non-descendants given its parents. The mask representation in (E.1) and (E.3) is equivalent to the concept of a structural gate used by Kalainathan et al. [63]. Note that the conditional distribution parameters for a single nonlinear Gaussian BN of the form in (E.3) contain the weights and biases of $d$ different neural networks, one for the local conditional distribution of each node.

## E.2   Evaluation metrics

We provide additional details on the evaluation metrics used throughout the paper. Bayesian structure learning beyond five variables is notoriously difficult to evaluate since the ground truth posterior is not accessible. We hence rely and build on the metrics established in the literature.

**Expected structural Hamming distance**   The structural Hamming distance $\mathrm{SHD}(\mathbf{G}, \mathbf{G}^*)$ between two graphs $\mathbf{G}$ and $\mathbf{G}^*$ counts the edge changes that separate the essential graphs representing the MECs of $\mathbf{G}$ and $\mathbf{G}^*$ [8, 55]. We define the expected structural Hamming distance to the ground truth graph $\mathbf{G}^*$ under the inferred posterior as

$$\mathbb{E}\text{-SHD}(p, \mathbf{G}^*) := \sum_{\mathbf{G}} p(\mathbf{G} \,|\, \mathcal{D}) \cdot \mathrm{SHD}(\mathbf{G}, \mathbf{G}^*)\ .$$

Empirically, the $\mathbb{E}$-SHD is similar to the $L_1$ edge error used by Tong and Koller [11] and Murphy [10], but also takes into account the MEC. The $\mathbb{E}$-SHD is computed via Monte Carlo estimation of the expectation using samples from the posterior. Note that the DAG bootstrap variants and DiBS+ use the weighted mixture rather than the empirical distribution of samples. In the joint inference setting, we empirically marginalize out $\mathbf{\Theta}$ to obtain $p(\mathbf{G} \,|\, \mathcal{D})$.

**Receiver operating characteristic**   The marginal posterior $p(\mathbf{G} \,|\, \mathcal{D})$ provides a confidence estimate $p(g_{ij} = 1 \,|\, \mathcal{D})$ for whether a given edge $(i, j)$ is present in the ground truth DAG $\mathbf{G}^*$. Recall that the marginal posterior edge probability $p(g_{ij} = 1 \,|\, \mathcal{D})$ is the posterior mean of an indicator for the presence of that edge, i.e., $p(g_{ij} = 1 \,|\, \mathcal{D}) = \mathbb{E}_{p(\mathbf{G} \,|\, \mathcal{D})} \mathbb{1}[g_{ij} = 1]$, which amounts to counting the proportion of graphs with $g_{ij} = 1$ (and to weighted counting for the DAG bootstrap variants and DiBS+). The receiver operating characteristic (ROC) curve is then obtained by viewing the presence of each of the $d^2$ possible edges in a $d$-node graph as a binary classification task and varying the decision threshold from 0 to 1 under our confidence estimates $p(g_{ij} = 1 \,|\, \mathcal{D})$. The area under the receiver operating characteristic curve (AUROC) evaluates faithful uncertainty quantification of the posterior. In general, random guessing achieves an AUROC of 0.5 in expectation; a perfect classifier achieves an AUROC of 1.

**Held-out log likelihood**   We also evaluate the ability to predict future observations by computing the average negative log likelihood on 100 held-out observations $\mathcal{D}^{\text{test}}$ defined as

$$\text{neg. LL}(p, \mathcal{D}^{\text{test}}) := -\sum_{\mathbf{G}, \mathbf{\Theta}} p(\mathbf{G}, \mathbf{\Theta} \,|\, \mathcal{D}) \cdot \log p(\mathcal{D}^{\text{test}} \,|\, \mathbf{G}, \mathbf{\Theta})\ .$$

As for $\mathbb{E}$-SHD, the neg. LL is a posterior mean and thus computed via Monte Carlo estimation using samples from the posterior. When inferring $p(\mathbf{G} \,|\, \mathcal{D})$, the corresponding neg. MLL metric uses the marginal likelihood $p(\mathcal{D}^{\text{test}} \,|\, \mathbf{G})$ instead of the likelihood $p(\mathcal{D}^{\text{test}} \,|\, \mathbf{G}, \mathbf{\Theta})$.

**Held-out log interventional likelihood**   Lastly, to capture relevant performance metrics in causal inference [10, 12], we also compute the negative *interventional* log likelihood. Given an interventional

data set $(\mathcal{D}^{\text{int}}, \mathcal{I})$, the interventional likelihood is given by

$$p(\mathcal{D}^{\text{int}} \mid \mathbf{G}, \mathbf{\Theta}, \mathcal{I}) = \prod_{\mathbf{x}^{(n)} \in \mathcal{D}} \prod_{\substack{j=1 \\ j \notin \mathcal{I}}} p(x_j^{(n)} \mid \mathbf{x}_{\mathbf{G}_j}^{(n)}, \boldsymbol{\theta}_j) \tag{E.4}$$

where $\mathbf{x}_{\mathbf{G}_j}$ are the values of the parents of variable $j$ in $\mathbf{G}$, and $\boldsymbol{\theta}_j$ parameterizes the local conditional distribution of $j$. The neg. I-LL and neg. I-MLL metrics are defined analogous to the neg. LL and neg. MLL in (19) but use the interventional likelihood in (E.4) instead of the observational likelihood. For marginal posterior inference, we likewise use the interventional marginal likelihood $p(\mathcal{D}^{\text{int}} \mid \mathbf{G}, \mathcal{I})$ instead. In our experiments, we obtain interventional data $(\mathcal{D}^{\text{int}}, \mathcal{I})$ by randomly selecting 10% of the variables and clamping them to zero in the ground-truth data-generating process. The reported neg. I-LL and I-MLL scores are the average of 10 different interventional data sets with $|\mathcal{D}^{\text{int}}| = 100$.

### E.3 Hyperparameters

In all evaluations, DiBS is run for 3,000 iterations and uses the simple linear constraint schedule $\beta_t := t$. At $t = 0$, the initial latent particles $\{\mathbf{Z}_0\}_{m=1}^M$ and parameter particles $\{\mathbf{\Theta}_0\}_{m=1}^M$ are initialized by sampling from their prior distributions. For the step size schedule $\eta_t$, we use the adaptive learning rate method RMSProp with learning rate 0.005. We always use 128 samples for Monte Carlo estimation of the gradients. Finally, the bandwidths $\gamma_z, \gamma_\theta$ of the kernel in (15) and the slope of a linear schedule $\alpha_t$ are chosen in separate held-out instances of each setting in Section 6 and are listed in Table 2. As illustrated by the application in Section 7, the provided hyperparameters can be expected to apply to problem settings of comparable magnitude.

While the latent variable scale $\sigma_z$ can in principle be set arbitrarily, we always set $\sigma_z = 1/\sqrt{k}$ in the prior $p(\mathbf{Z})$, which makes the norm in the SE kernel given in (15) roughly invariant with respect to the latent dimension $k$, ignoring the acyclicity term. This follows from the fact that $||\mathbf{u}||^2 \sim \text{Gamma}(k/2, 2\sigma_z^2)$ when $u_i \sim \mathcal{N}(0, \sigma_z^2)$, in which case $\mathbb{E}[||\mathbf{u}||^2] = k\sigma_z^2$.

Table 2: DiBS hyperparameter choices for $\alpha_t$ and bandwidths $\gamma_z, \gamma_\theta$. Here, $\widetilde{\alpha}$ denotes the slope in the linear schedule $\alpha_t := \widetilde{\alpha} t$.

| Model | $d$ | $\widetilde{\alpha}$ | $\gamma_z$ | $\gamma_\theta$ |
|---|---|---|---|---|
| BGe | 20 | 2 | 2 | - |
|  | 50 | 2 | 50 | - |
| Linear Gaussian | 20 | 0.2 | 5 | 500 |
|  | 50 | 0.02 | 15 | 1,000 |
| Nonlinear Gaussian | 20 | 0.02 | 5 | 1,000 |
|  | 50 | 0.01 | 15 | 2,000 |

### E.4 Baseline Methods

**Structure MCMC (MC$^3$, M-MC$^3$, G-MC$^3$)** Designed for inference of the marginal posterior $p(\mathbf{G} \mid \mathcal{D})$, structure MCMC [36, 37] performs sampling in the space of DAGs by adding, deleting, and reversing one edge at a time without violating acyclicity. The acceptance probability of a proposed graph $\mathbf{G}'$ is given by

$$\min \left\{ 1, \frac{|\mathcal{N}(\mathbf{G})| \cdot p(\mathcal{D} \mid \mathbf{G}') p(\mathbf{G}')}{|\mathcal{N}(\mathbf{G}')| \cdot p(\mathcal{D} \mid \mathbf{G}) p(\mathbf{G})} \right\} \tag{E.5}$$

where $\mathbf{G}$ is the current particle and $\mathcal{N}(\mathbf{G})$ is the collection of DAGs reachable from $\mathbf{G}$ with one edge change. Following [37], the ratio of neighborhoods is approximated to equal one, which allows for only computing $\mathcal{N}(\mathbf{G}')$ when accepting $\mathbf{G}'$. We implement MC$^3$ using the efficient ancestor matrix trick for finding acyclic proposals [37]. For marginal inference under the BGe marginal likelihood, we compute the Bayes factor in (E.5) by only taking into account the affected node families.

For all of MC$^3$, M-MC$^3$, and G-MC$^3$, we specify a burn-in period of 100k samples and then collect a sample every 10k steps, which makes the wall time of MC$^3$ and DiBS on CPUs comparable. Both M-MC$^3$ and G-MC$^3$ use a simple Gaussian random walk proposal for the parameters, respectively, with scale selected to roughly obtain an acceptance rate of 0.2 in each setting [64], when feasible in combination with the graph proposal.

**Nonparametric DAG bootstrap (BPC, BGES, BPC$^*$, BGES$^*$)**   The nonparametric DAG bootstrap [42] performs model averaging by bootstrapping the observations $\mathcal{D}$ to yield a collection of synthetic data sets, each of which is used to learn a single graph, here using the GES and PC algorithms [6, 7]. The collection of unique single graphs approximates the posterior by weighting each graph by its unnormalized posterior probability in (2), analogous to DiBS+. The closed-form maximum likelihood parameter estimate for linear Gaussian BNs with known $\mathbf{G}$, which is used by BPC$^*$ and BGES$^*$ to allow approximating the joint posterior, is provided by Hauser and Bühlmann [57]. For joint posterior inference, BPC$^*$ and BGES$^*$ use $p(\mathbf{G}, \mathbf{\Theta}, \mathcal{D})$ rather than $p(\mathbf{G}, \mathcal{D})$ for weighting the inferred BN models.

Since the GES and PC algorithms only return essential graphs, i.e., MECs, we favor them in computing the AUROC score. We orient a predicted undirected edge correctly when a ground truth edge exists and only count a falsely predicted undirected edge as a single mistake. The held-out likelihood metrics given in (19) are computed for a random consistent DAG extension of the essential graph [65]. Enumerating the possibly exponential number of DAGs in an MEC is infeasible in general [66]. Implementations of the PC and GES algorithms are given by the `CausalDiscoveryToolbox` [67], which is published under an MIT Licence and executes their commonly used R implementations.

# F   Efficient Implementation and Computing Resources

DiBS and SVGD operate on continuous tensors and solely rely on Monte Carlo estimation and parallel gradient ascent-like updates. Thus, our inference framework allows for a highly efficient implementation using vectorized operations, automatic differentiation, just-in-time compilation, and hardware acceleration. For this purpose, we implement DiBS with `JAX` [68], which is published under an Apache Licence. Our code is publicly available at: https://github.com/larslorch/dibs.

Table 3 summarizes the computing time of DiBS on GPU and CPUs for a superset of the inference tasks on BNs with Erdős-Rényi structures in Section 6. The GPU wall times for medium-sized inference tasks of up to around 20 nodes and 30 particles lie on the order of seconds or a few minutes. None of the baseline methods considered in this work are comparable in terms of efficiency and usage of modern hardware accelerators. Moreover, in larger inference problems than evaluated here, `JAX` would directly allow for the computations of DiBS to be performed in a distributed fashion, e.g., by updating batches of SVGD particles across multiple GPU devices. Note that BGe wall times are relatively slow because the closed-form marginal likelihood involves determinants [16, 17].

Table 3: Wall times of DiBS for the hyperparameters described in Section E.3. Times are the mean of 10 random restarts. $M$ denotes the number of particles, $d$ the number of nodes, and GPU/CPU the processing backend of `JAX`. We used one NVIDIA GEFORCE RTX 2080 TI GPU or one full 2.70GHz INTEL XEONGOLD 6150 CPU node to measure GPU and CPU wall time, respectively, for each run. Experiments marked by a dash exceeded the GPU memory. The main experiments of Sections 6 and 7 were performed in bulk on Oracle BM.STANDARD.E2.64 CPU machines and no GPUs.

| | | | Wall time (min) | | | | |
| Model | | | GPU | | | CPU | |
| $d =$ | 10 | 20 | 50 | 10 | 20 | 50 |
|---|---|---|---|---|---|---|
| **BGe** | | | | | | |
| $M = 10$ | 0.349 | 0.892 | 8.111 | 3.337 | 18.251 | 216.844 |
| $M = 30$ | 0.609 | 2.084 | — | 9.380 | 52.203 | 659.106 |
| **Linear Gaussian** | | | | | | |
| $M = 10$ | 0.370 | 0.603 | 1.571 | 1.410 | 3.738 | 21.531 |
| $M = 30$ | 0.612 | 1.153 | 3.975 | 4.703 | 13.163 | 74.914 |
| **Nonlinear Gaussian** | | | | | | |
| $M = 10$ | 2.702 | 6.139 | 24.117 | 10.019 | 27.749 | 128.820 |
| $M = 30$ | 7.476 | 17.667 | — | 28.130 | 79.628 | 388.992 |

# G   Results for Gaussian Bayesian networks with $d = 50$ variables

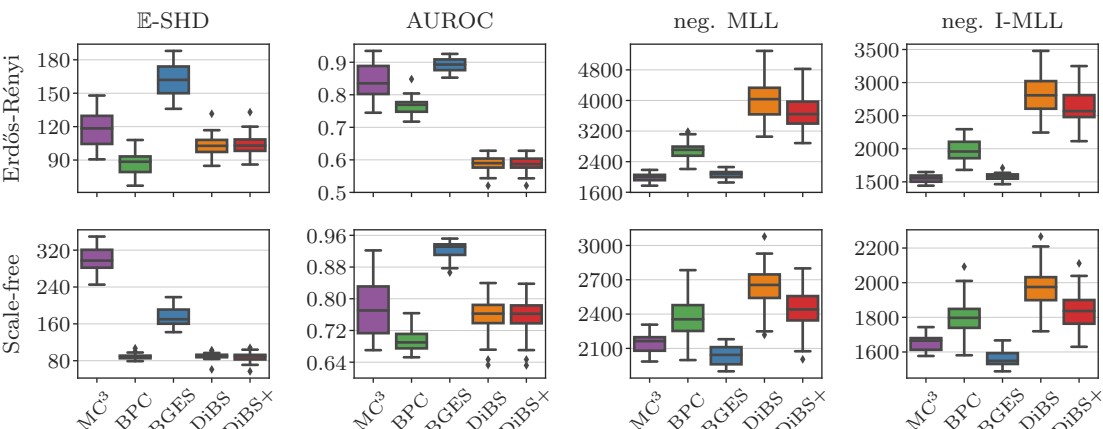

Figure 5: Marginal posterior inference of linear Gaussian BNs with $d = 50$ variables using the BGe marginal likelihood. The metrics are aggregated for 30 random BNs of each graph type. While DiBS and DiBS+ are competitive in the structural $\mathbb{E}$-SHD and AUROC metrics, we find that the baselines specifically designed for marginal posterior inference perform favorably in the likelihood-based metrics. We hypothesize that this is due to the high variance incurred by the score function estimator that DiBS needs to use in the marginal inference setting under the BGe model (cf. Section 6.2). To reach comparable results with DiBS in this high-dimensional setting, the DiBS score function gradient estimator may require more than the default 128 Monte Carlo samples used here.

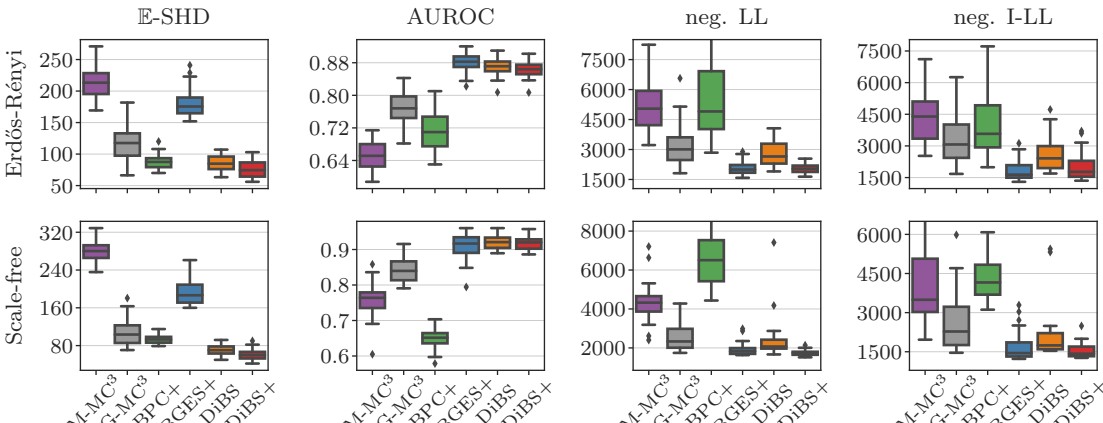

Figure 6: Joint posterior inference of linear Gaussian BNs with $d = 50$ variables. The first and second rows show the aggregate metrics for inference of 30 random BNs with Erdős-Rényi and scale-free structures, respectively. Analogous to inference for $d = 20$ variables, DiBS+ outperforms all alternatives to joint posterior inference of the graph and the conditional distribution parameters across the metrics.

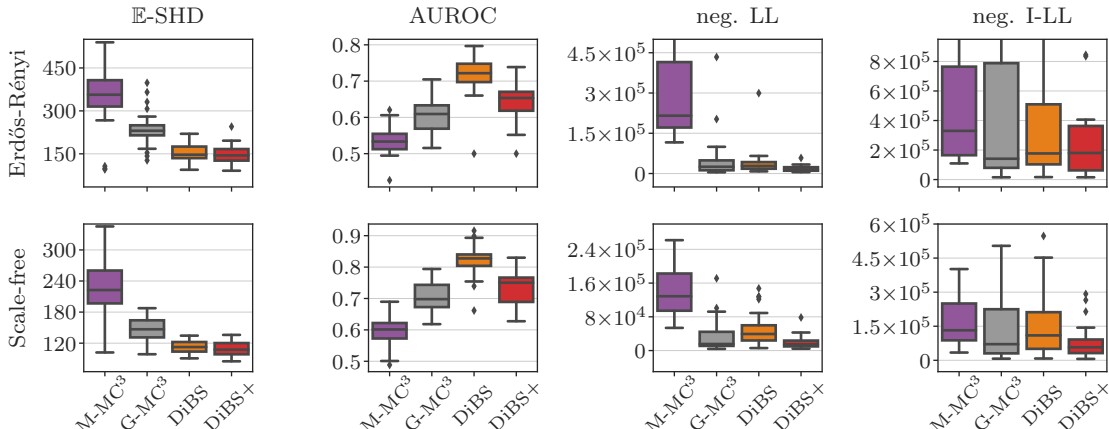

Figure 7: Joint posterior inference of nonlinear Gaussian BNs with $d = 50$ variables, where each local conditional distribution is parameterized by a 2-layer neural network with five hidden nodes. In this setting, the total number of conditional distribution parameters in a given BN amounts to $|\Theta| = 13{,}050$ weights and biases. The metrics are aggregated for inference of 30 random BNs of each graph type. Here, DiBS only infers 10 particles to make the wall time on CPUs comparable to M-MC[3] and G-MC[3]. As for posterior inference of BNs with $d = 20$ variables, DiBS and DiBS+ perform favorably compared to the MC[3] baselines.

## H    Additional Analyses and Ablation Studies

Having compared DiBS with several alternative approaches to Bayesian structure learning in Section 6, this supplementary section is devoted to a more in depth analysis of some of its properties. This is done by changing, or leaving out single design aspects of the algorithm and studying the effect on the previous metrics.

As in Section 6, DiBS and its instantiation with SVGD are used interchangeably here, and DiBS+ denotes the weighted mixture of particles. Since the metrics do not qualitatively differ between inference of Erdős-Rényi and scale-free BN structures in our experiments of Section 6, we only consider the former here. Unless mentioned otherwise, the following experimental setup corresponds to *joint* posterior inference of linear Gaussian BNs with $d = 20$ variables in Section 6.2.

### H.1    Graph Embedding Representation

In Section 4.2, we propose to use a generative graph model $p_\alpha(\mathbf{G} \mid \mathbf{Z})$ that is based on the inner product of latent embeddings for each node. In particular, we choose $p_\alpha(g_{ij} = 1 \mid \mathbf{Z}) = \sigma_\alpha(\mathbf{u}_i^\top \mathbf{v}_j)$ with latent variables $\mathbf{Z} = [\mathbf{U}, \mathbf{V}]$. In Figure 8, we contrast this modeling choice with the more trivial variant $p_\alpha(g_{ij} = 1 \mid \mathbf{Z}) = \sigma_\alpha(\mathbf{z}_{ij})$, where single *scalars* rather than inner products between latent

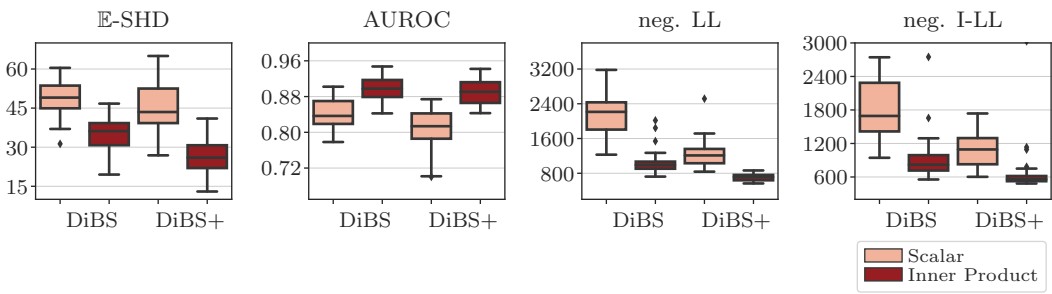

Figure 8: Contrasting the bilinear graph model of Section 4.2 with its more trivial variant, where each latent variable models the edge probabilities directly via the sigmoid. The plots aggregate the results for joint inference of 30 randomly generated linear Gaussian BNs with $d = 20$ variables.

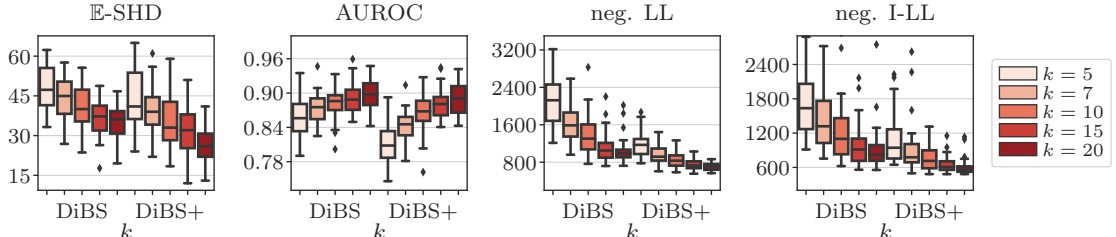

Figure 9: DiBS for joint inference of linear Gaussian BNs with $d = 20$ variables for different sizes of the latent variables $\mathbf{Z} \in \mathbb{R}^{2 \times d \times k}$. Lower rank parameterizations of the matrix of edge probabilities balance the tradeoff between computational efficiency and posterior approximation quality.

vectors encode the edge probabilities. Bengio et al. [69] and Ke et al. [31] use the scalar variant with fixed $\alpha = 1$ in the context of causal inference.

The comparison in Figure 8 illustrates that incorporating only the bilinear parameterization of edge probabilities in the generative graph model improves performance by a significant margin. We hypothesize that the coupling between edges results in smoother densities, which might be less prone to local minima in gradient-based methods such as DiBS.

## H.2   Graph Embedding Dimensionality

Another feature of the inner product representation of graphs is the ability to control the dimensionality of the posterior inference task. As described in Section 5, we generally set $k = d$ for the latent variables $\mathbf{Z} \in \mathbb{R}^{2 \times d \times k}$ that parameterize our graph model $p_\alpha(\mathbf{G} \,|\, \mathbf{Z})$. This leaves the matrix of edge probabilities fully expressible and without a rank constraint. In principle, however, the formulation in (6) allows us to arbitrarily vary $k$. This creates a trade-off between the complexity of the parameterization and the tractability and dimensionality challenges in approximate inference of $p(\mathbf{Z} \,|\, \mathcal{D})$ or $p(\mathbf{Z}, \mathbf{\Theta} \,|\, \mathcal{D})$. Limiting $k < d$ has connections to the theory of low-rank realizations of sign matrices.

We perform inference with DiBS for $k \in \{5, 7, 10, 15, 20\}$, leaving all other aspects of the algorithm unchanged. Hence, the corresponding posterior over $\mathbf{Z}$ has $\{200, 280, 400, 600, 800\}$ dimensions, respectively. The results in Figure 9 suggest that lower values of $k = 15$, or even $k = 10$, are already able to achieve competitive performance across all metrics. Interestingly, the structural $\mathbb{E}$-SHD metric appears to suffer most from a small loss in complexity.

In this context, one should keep in mind that the bandwidth parameters $\gamma_z$ and $\gamma_\theta$ were set to achieve good performance with $k = d = 20$. It is possible that lower values of $k$ can reach performances that are even closer to the full-rank variant of DiBS with alternative settings for $\gamma_z$ and $\gamma_\theta$. In addition, a lower-rank DiBS variant could be particularly promising for inference of very large BNs, where the

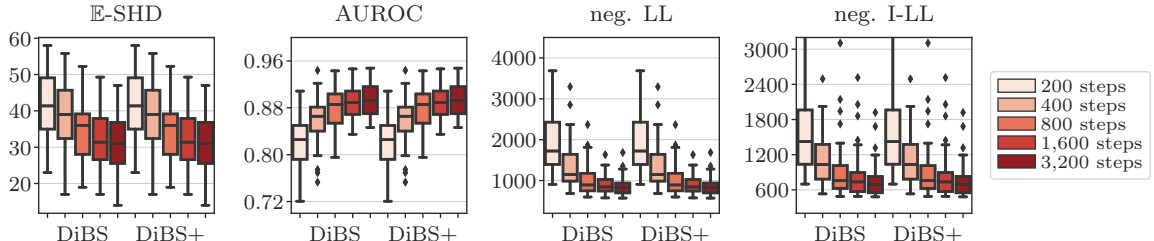

Figure 10: Performance of DiBS and DiBS+ as a function the number of particle transport steps $T$. As previously, the plots aggregate the results for inference of 30 randomly generated linear 20-node Gaussian BNs. The latent variable $\mathbf{Z}$ is specified with its default dimensions $k = d = 20$. After already roughly 1,000 iterations, DiBS and DiBS+ obtain good posterior approximations.

computational challenges of the full $O(d^2)$ latent representation might outweigh its benefits in terms of expressibility.

## H.3 Particle Transport Iterations

Since DiBS uses Stein variational gradient descent [19] for posterior inference, our method iteratively transports a set of latent graph particles, or latent graph and parameter tuples, for a number of $T$ steps. As the particles are randomly initialized, the approximation quality of SVGD particles (and thus also DiBS particles) improves with the number of steps. We are interested in the degree to which a small number of transport steps provide a good posterior approximation in the face of computational constraints.

In Figure 10, we show the performance of DiBS as a function of the number of transport steps. We find that even a smaller number of iterations achieves competitive results across the metrics. In addition, the variance of performance in the predictive metrics neg. LL and neg. I-LL decreases monotonically as a function of the performed transforms, whereas the variation in $\mathbb{E}$-SHD remains roughly the same.

## H.4 Uncertainty Quantification Within a Markov Equivalence Class

When performing posterior inference of $p(\mathbf{G} \mid \mathcal{D})$ with the BGe marginal likelihood and a uniform prior $p(\mathbf{G})$, each Markov equivalent structure is assigned equal likelihood. This might be desirable considering that causal edge directions are often not fully identifiable from purely observational data. As DiBS infers a posterior over DAGs rather than MECs, we aim to validate the ability of DiBS to correctly quantify the uncertainty present in nonidentifiable edge directions.

To this end, we consider a 4-node example Bayesian network, small enough to allow for the closed-form computation of the ground-truth posterior by exhaustive enumeration of all possible DAGs. This enables us to compute the true single and pairwise posterior edge marginals and contrast them with the approximate posterior marginals inferred by DiBS. The graph structure for this analysis is chosen to contain both an identifiable v-structure and a nonidentifiable edge pair. Figure 11 shows the ground truth DAG $\mathbf{G}_0^*$, its linear Gaussian parameters, and the observational model. In addition,

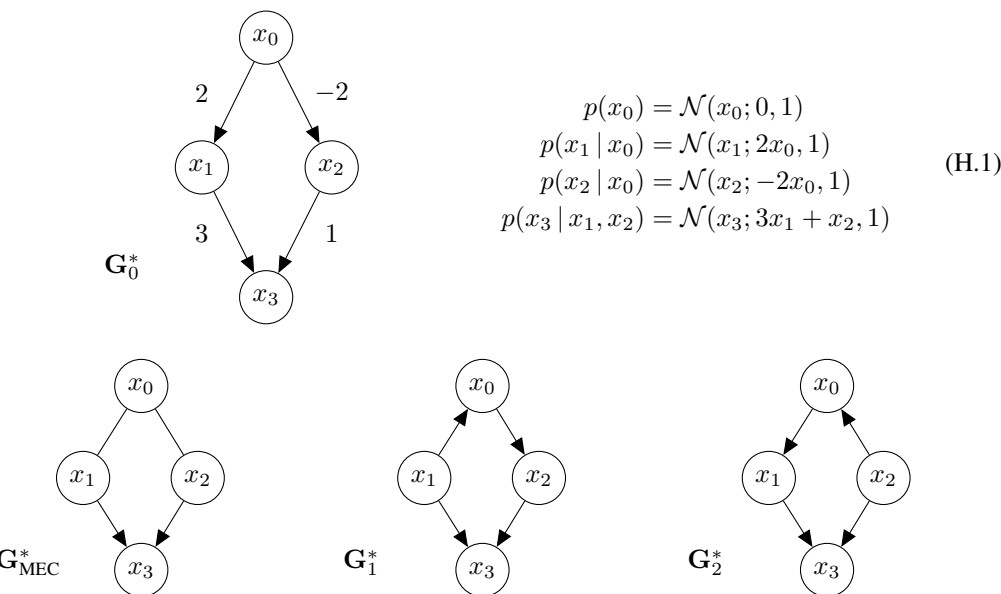

$$
\begin{aligned}
p(x_0) &= \mathcal{N}(x_0; 0, 1) \\
p(x_1 \mid x_0) &= \mathcal{N}(x_1; 2x_0, 1) \\
p(x_2 \mid x_0) &= \mathcal{N}(x_2; -2x_0, 1) \\
p(x_3 \mid x_1, x_2) &= \mathcal{N}(x_3; 3x_1 + x_2, 1)
\end{aligned}
\tag{H.1}
$$

Figure 11: Four-node example linear Gaussian Bayesian network. Under the BGe marginal likelihood and a uniform prior, $\mathbf{G}_0^*$, $\mathbf{G}_1^*$, and $\mathbf{G}_2^*$ are scored equally. While the v-structure $x_1 \rightarrow x_3 \leftarrow x_2$ is an identifiable feature of the MEC of $\mathbf{G}_0^*$ and thus present in $\mathbf{G}_{\text{MEC}}^*$, the edge directions of $x_1 \,\text{---}\, x_0 \,\text{---}\, x_2$ cannot be distinguished even given infinite observational data.

Table 4: Ground truth and average inferred posterior marginals given $N = 100$ observations from the ground truth model in Figure 11. Listed are the probabilities for the nonidentifiable edge structure $x_1 - x_0 - x_2$ (top) and the identifiable v-structure $x_1 \rightarrow x_3 \leftarrow x_2$ (bottom). Averaged over 30 random particle initilizations, DiBS+ correctly quantifies the confidence and uncertainty in the v-structure and nonidentifable edge pair, respectively.

| | DiBS | DiBS+ | **Ground Truth** |
|---|---|---|---|
| $p(x_1 \rightarrow x_0, x_0 \rightarrow x_2 \mid \mathcal{D})$ | 0.134 | 0.216 | **0.298** |
| $p(x_1 \rightarrow x_0, x_0 \leftarrow x_2 \mid \mathcal{D})$ | 0.201 | 0.037 | **0.052** |
| $p(x_1 \leftarrow x_0, x_0 \rightarrow x_2 \mid \mathcal{D})$ | 0.223 | 0.409 | **0.311** |
| $p(x_1 \leftarrow x_0, x_0 \leftarrow x_2 \mid \mathcal{D})$ | 0.103 | 0.298 | **0.297** |
| $p(x_1 \rightarrow x_3, x_3 \rightarrow x_2 \mid \mathcal{D})$ | 0.157 | 0.013 | **0.017** |
| $p(x_1 \rightarrow x_3, x_3 \leftarrow x_2 \mid \mathcal{D})$ | 0.338 | 0.934 | **0.914** |
| $p(x_1 \leftarrow x_3, x_3 \rightarrow x_2 \mid \mathcal{D})$ | 0.275 | 0.034 | **0.043** |
| $p(x_1 \leftarrow x_3, x_3 \leftarrow x_2 \mid \mathcal{D})$ | 0.154 | 0.019 | **0.025** |

Figure 11 lists the essential graph $\mathbf{G}^*_{\text{MEC}}$ as well as the two other Markov equivalent DAGs $\mathbf{G}^*_1$ and $\mathbf{G}^*_2$ in the MEC represented by $\mathbf{G}^*_{\text{MEC}}$.

We perform marginal posterior inference with DiBS using the experimental setup and hyperparameters for 20-node linear Gaussian BNs. In this example setting, DiBS employs a uniform prior over graphs and uses the default $k = d = 4$. Table 4 shows the ground truth and inferred pairwise edge marginals under the posterior. We find that DiBS+ correctly infers both the uncertainty in the edge directions of $x_1 - x_0 - x_2$ as well as the high confidence in the presence of the v-structure $x_1 \rightarrow x_3 \leftarrow x_2$. While the unweighted particles of DiBS do not exhibit false confidence in structures that are not present in $\mathbf{G}^*_0$, its inferred degree of uncertainty is too high compared to the ground truth. The DiBS+ variant overcomes the inexact empirical average of DiBS by weighting the particles by their unnormalized posterior probabilities.

# I  Experimental Details for Application to Protein Signaling Networks

The data by Sachs et al. [3] as well as the corresponding consensus graph used in Section 7 are taken as provided by the `CausalDiscoveryToolbox` [67], which is published under an MIT Licence. We standardize the data for inference. Because $N$ is large, DiBS uses minibatches of 100 observations to estimate the scores of the posterior. All hyperparameters and BN specifications are chosen by default exactly as during the synthetic evaluation of linear and nonlinear Gaussian BNs in Table 2, respectively, except that DiBS correspondingly uses $k = d = 11$. For joint posterior inference of linear Gaussian BNs, BPC* and BGES* still use the BGe marginal likelihood; since metrics are very similar to BPC and BGES, their scores are not reported.

In line with inference on synthetic data in Section 6, the BGe marginal likelihood employed for the experiments in Section 7 uses the default effective sample sizes $\alpha_\mu = 1$ and $\alpha_\omega = d + 2$ described in Appendix E.1. Likewise, we again set the noise level for inference with the explicitly parameterized linear and nonlinear Gaussian networks to $\sigma^2 = 0.1$.

Since the effective sample size $\alpha_\mu$ and the noise level $\sigma^2$ may affect the model complexity of the inferred BNs, e.g., the mean number of inferred edges in the DAG, we provide additional results for alternative values of these Bayesian network model hyperparameters in Tables 5 and 6. Overall, we find that increasing the effective sample size $\alpha_\mu$ does not significantly change metrics across the considered methods. However, higher fixed noise levels $\sigma^2$ do result in less inferred edges, which tends to lead to lower $\mathbb{E}$-SHD but worse AUROC, i.e., less calibrated edge confidence scores. We note that these are not free parameters of the inference methods that approximate the posterior, but specifications of the inferred BN models themselves.

Table 5: Additional results for marginal posterior inference of protein signaling pathways under the BGe marginal likelihood of linear Gaussian BNs [16, 17]. Changing the effective sample size in the BGe Normal-Wishart prior does not result in significantly different metrics compared to the default $\alpha_\mu = 1$ used in all of our experiments. For $\alpha_\mu = 10$, DiBS and DiBS+ average an expected number of 39.6 and 35.4 edges, respectively. Metrics are the mean $\pm$ SD of 30 random restarts.

| | $\alpha_\mu = 10$ | |
| | $\mathbb{E}$-**SHD** | **AUROC** |
|---|---|---|
| MC$^3$ | $34.3 \pm 0.4$ | $0.622 \pm 0.020$ |
| BPC | $25.5 \pm 2.3$ | $0.566 \pm 0.020$ |
| BGES | $33.8 \pm 1.8$ | $0.641 \pm 0.034$ |
| DiBS | $37.9 \pm 0.5$ | $0.637 \pm 0.046$ |
| DiBS+ | $35.1 \pm 1.8$ | $0.627 \pm 0.050$ |

Table 6: Additional results for joint posterior inference of protein signaling pathways under explicitly parameterized linear (top) and nonlinear (bottom) Gaussian BNs. The hyperparameter $\sigma^2$ specifies the noise level underlying the inferred Bayesian networks. We find that higher noise levels $\sigma^2$ tend to result in less edges. When inferring linear Gaussian BNs with $\sigma^2 = 0.01$ ($\sigma^2 = 1$), DiBS averages an expected number of 11.6 (8.8) edges, DiBS+ 13.8 (9.9) edges. For nonlinear Gaussian BNs with $\sigma^2 = 0.01$ ($\sigma^2 = 1$), DiBS averages 15.6 (5.2) edges, DiBS+ 17.5 (6.8) edges. Due to less false positives, the $\mathbb{E}$-SHD improves, but the degree of uncertainty in the presence of edges is quantified less accurately, resulting in worse AUROC. Metrics are the mean $\pm$ SD of 30 random restarts.

| | $\sigma^2 = 0.01$ | | $\sigma^2 = 1$ | |
| | $\mathbb{E}$-**SHD** | **AUROC** | $\mathbb{E}$-**SHD** | **AUROC** |
|---|---|---|---|---|
| M-MC$^3$ | $38.1 \pm 3.4$ | $0.536 \pm 0.082$ | $33.1 \pm 3.4$ | $0.543 \pm 0.105$ |
| G-MC$^3$ | $30.9 \pm 3.0$ | $0.518 \pm 0.051$ | $29.8 \pm 3.7$ | $0.531 \pm 0.078$ |
| DiBS | $23.0 \pm 0.5$ | $0.595 \pm 0.069$ | $20.3 \pm 0.4$ | $0.601 \pm 0.039$ |
| DiBS+ | $22.9 \pm 2.0$ | $0.540 \pm 0.048$ | $20.0 \pm 1.4$ | $0.569 \pm 0.040$ |

| | $\sigma^2 = 0.01$ | | $\sigma^2 = 1$ | |
| | $\mathbb{E}$-**SHD** | **AUROC** | $\mathbb{E}$-**SHD** | **AUROC** |
|---|---|---|---|---|
| M-MC$^3$ | $38.7 \pm 3.2$ | $0.555 \pm 0.101$ | $18.4 \pm 0.1$ | $0.501 \pm 0.043$ |
| G-MC$^3$ | $34.9 \pm 3.6$ | $0.542 \pm 0.064$ | $30.6 \pm 2.5$ | $0.538 \pm 0.059$ |
| DiBS | $24.3 \pm 0.6$ | $0.582 \pm 0.050$ | $17.7 \pm 0.1$ | $0.550 \pm 0.020$ |
| DiBS+ | $24.9 \pm 2.9$ | $0.535 \pm 0.045$ | $18.5 \pm 0.5$ | $0.530 \pm 0.028$ |