# OpenReview forum: "DiBS: Differentiable Bayesian Structure Learning"
_NeurIPS.cc/2021/Conference — NeurIPS 2021 Spotlight_

### Official Review · Reviewer_ZLxN · 2021-07-16

**Rating:** 8
**Confidence:** 3

**Summary:**

This paper proposes a method called DiBS (differentiable Bayesian structure learning) for learning the structure of a Bayesian network (BN) from data based on a fully differentiable reparameterization of the relevant likelihood expressions, allowing for the use of continuous optimization-based Bayesian inference algorithms. This method allows for posterior inference of graph structure alone, or joint inference over structure and conditional distribution parameters. First, they augment the standard generative model of data from a Bayesian network with a continuous-valued latent variable of node embeddings such that the probability of each edge in the graph adjacency matrix can be modeled by the inner product similarity of node embeddings with a temperature parameter. The authors use a previously-introduced function of the adjacency matrix that calculates graph cyclicity to include a prior that encourages acyclic structure. They derive expressions that reformulate the necessary score functions in terms of gradients with respect to the new continuous latent graph representation, using the Gumbel-softmax trick to reparameterize the gradient where applicable and the score function gradient estimator otherwise. The method uses Stein variational gradient descent as the inference algorithm, optimizing a set of particles for the latent graph representation while annealing the temperature parameters to result in a final set of particles that correspond to adjacency matrices of valid directed acyclic graphs. A variant of the base method called DiBS+ that weights each particle by its unnormalized posterior probability is also explored. On synthetic data from linear Gaussian BNs, DiBS and DiBS+ show comparable performance to greedy search-based (BPC, BGES) and sampling-based ($\text{MC}^3$) algorithms across a variety of metrics when inferring structure alone, and comparable or better performance when jointly inferring structure and parameters. On synthetic data with nonlinear local conditionals parameterized by a neural network, DiBS/DiBS+ perform comparably to or better than sampling-based methods for joint inference across metrics. These results are also reflected on real data from a protein signaling network, where DiBS/DiBS+ achieve the best recovery of the true network and the most well-calibrated edge confidence scores in the joint inference setting.

**Ethical Concerns:**

No ethical concerns come to mind.

**Limitations And Societal Impact:**

The authors briefly discuss the potential negative impacts, specifically the risk in taking a causal interpretation of the results of this method applied to real data without considering the necessary assumptions.

**Main Review:**

This paper presents a very interesting approach for joint Bayesian inference of structure and parameters of graphical models using continuous optimization-based methods, with the overall presentation of the method being clear with sufficient detail. While the performance of DiBS is comparable to other methods for marginal posterior inference in the linear Gaussian setting, it still provides a competitive alternative to existing approaches. DiBS seems to particularly excel when applied to joint inference of structure and parameters, especially with nonlinear conditionals, which is relevant for a lot of real-world applications and presents this method as a potentially valuable tool for practitioners interested in this kind of inference for their data. Given that DiBS is also agnostic to the particular inference algorithm used (as the authors mention in their Conclusion), any advancements in Bayesian inference algorithms that require a differentiable likelihood could be applied to improve upon this framework. Overall, I believe this is a strong paper and recommend it for acceptance.

**Time Spent Reviewing:**

2.5

---

> ### Author Response · Authors · 2021-08-09
> **Response to Reviewer ZLxN**
>
> Thank you for your assessment and interest in our work. We are pleased to hear that you find our contribution clear and interesting. We will be happy to clarify further questions in case they arise.

---

### Official Review · Reviewer_GxXy · 2021-07-16

**Rating:** 7
**Confidence:** 3

**Summary:**

This work proposes a new approach to posterior approximation in the context of structure learning based on the Stein Variational Gradient Descent (SVGD) procedure and the differentiable acyclicity constraint inspired from Zheng et al 2018. A continuous latent variable Z (parent of G only) is introduced in the model to allow the computation of the gradient of its posterior (with respect to Z) which in turn enables one to use SVGD to sample from P(G|D). This gradient can be estimated using the Gumbel-softmax trick or score function estimator. The framework is validated on synthetic data as well as the protein signaling dataset from Sachs et al..

**Limitations And Societal Impact:**

Yes

**Main Review:**

Originality:

The method proposed is original, however, I believe two papers strongly related to this work have been omitted, namely, “Masked Gradient-Based Causal Structure Learning” by Ng et al. and “Differentiable Causal Discovery from Interventional Data” by Brouillard et al.. Both works leverage the Gumbel-softmax trick to learn an adjacency matrix (like this work), although without framing their approach as Bayesian (thus without learning a posterior distribution over graphs). The latter work frames its continuous constrained optimization problem as a stochastic relaxation of the original discrete combinatorial problem. Given the strong resemblance (when using a single particle), I believe it is important that the paper contrasts these approaches with theirs. An empirical comparison (with the single particle version) would also be interesting.

Quality:

The method proposed is well motivated and technically sound. I have not reviewed the proofs however. This is an interesting direction and I believe this a good paper. However, there are some things I would like the authors to clarify/adjust, mainly regarding experiments. I am open to revise my score once the following concerns are addressed:

L219: Could you provide some motivation for this “weighted particle mixture” modification? Does SVGD allow for this kind of modification or is it a contribution from the authors? This is especially important since it appears to be performing better in some cases.

I am under the impression that the metrics reported will favor methods which underestimate the uncertainty in the posteriors. For instance, if the samples from the approximate posterior (wrongfully) concentrate around the MAP estimator, E-SHD will be potentially lower than it would have been if the samples were coming from the actual posterior. I am not well-versed in Bayesian model selection so I don’t know what are the typical ways to evaluate the quality of the posterior, but I suspect it needs to somehow take into account proper uncertainty quantification. This aspect is studied in Appendix E.4 for a very special case of a graph with 4 nodes, allowing to compute the exact posterior and compare it with the one approximated by DiBS. I noticed that the number of SVGD iterations in this study is much smaller than in the experiment presented in the main paper (500 against 3000). I would like to see how the numbers from Table 4 change if one makes 3000 SVGD iterations. Does the uncertainty estimation remain the same? Given the importance that this work puts on being able to estimate p(G|D), I believe it would be important to make this kind of analysis on more graphs and more methods to see how DiBS compare. These experiments should also appear in the main text given their importance.

L268: Is 30 samples from the posterior enough to estimate the various performance metrics? It would be interesting to see how much the metrics vary when one repeats the sampling. Do the performance metrics change a lot when using more samples from the posterior?

Clarity:

The paper is well written and I enjoyed reading it. Here are a few things that might still be clarified:

In Section 4.3, titled “Black box bayesian inference”, I was not sure what the actual algorithm suggested was. I believe the title is a bit misleading since no specific inference procedure is suggested in this section. The first paragraph of the following section also implicitly mentions that some inference procedure was suggested above: “... translates learning discrete graph structures into an inference problem over the continuous variable Z.” As an outsider to the Bayesian approach, the only inference procedure I thought of after reading only 4.3 is *maximum a posteriori estimation* via gradient ascent. Later reading the paragraph at line 212 confirmed my guess. Please clarify.

Equation (15) felt a bit hard to interpret given that no explanation of the stein variational gradient descent was provided. Maybe adding some general explanation of this method in the background section would help, or maybe just refer to Algorithm 1 at the very beginning of Section 5.

Minor:

Equation (8): Why using E_{G | Z}[h(G)] instead of h(G_\alpha(Z)) ? After all, h was designed to take as input weighted adjacency matrices.

Equation (17) if <u_i, v_j> = 0, the limit is ½ right? Probably not relevant since this concerns a set of measure zero.

For each method, which thresholds were explored to compute the AUROC, and how were they selected? Also regarding the AUROC, how is p(g_ij = 1 | D) estimated? I assume by counting the proportion of graphs with g_ij = 1 among the 30 samples from the posterior?

Confused by ijk in equation (8), I thought Z was a k x 2d matrix (line 129)

How is E-SHD estimated? I am guessing via Monte Carlo estimation using the samples from the posterior. To clarify.

Significance:

I believe proper quantification of uncertainty in structure learning is important and this work makes an interesting contribution in that direction by using recent developments in continuous optimization for structure learning. This work has potential for impact.





**Time Spent Reviewing:**

5

---

> ### Author Response · Authors · 2021-08-09
> **Response to Reviewer GxXy**
>
> We thank you for your careful reading and assessment of our work.
>
> First, we appreciate the pointer to the two papers by Ng et al. and Brouillard et al. As related work in the realm of (non-Bayesian) structure learning, we followed your suggestion and added both references in lines 75-77 of Section 2. Other than also leveraging the continuous acyclicity penalizer [11, 25] and the widely-used Gumbel-softmax trick for modeling discrete random variables [47, 48], we currently do not see methodological overlap with DiBS beyond this. The key contribution of our work lies in representing and approximating full posterior distributions over Bayesian networks using a novel latent variable model, which is separate from the objectives of Ng et al. and Brouillard et al.
>
> In the following, we reply point-by-point to your more specific concerns:
> > L219: Could you provide some motivation for this “weighted particle mixture” modification? Does SVGD allow for this kind of modification or is it a contribution from the authors? [...]
>
> Overall, we motivate weighting the inferred particles in DiBS+ by the equivalent weighting procedure performed by the nonparametric DAG bootstrap [41]. In addition, note that while SVGD operates in the continuous space of $Z^{(m)}$ to approximate the posterior _density_ $p(Z|D)$, DiBS+ ultimately weighs the _discrete_ adjacency matrices $G^{(m)}$ to approximate the posterior $p(G|D)$, i.e., in the realm of probability mass functions rather than densities, beyond the density inferred by SVGD itself. However, we readily admit that we do not have a strong theoretical justification for the weighting at this point, and we will be happy to flag this in the paper.
>
> We hypothesize that the weighting of DiBS+ helps empirically because most of the particles in the empirical distribution of inferred graph particles are unique (which is a consequence of the vast, super-exponentially large space of DAGs). As a result, the DiBS distribution is nearly uniform over a few “likely” DAG candidates, and additionally weighting these by their likelihood may result in a closer approximation of the true posterior.
>
> > I am under the impression that the metrics reported will favor methods which underestimate the uncertainty in the posteriors. [...] This aspect is studied in Appendix E.4 for a very special case of a graph with 4 nodes, allowing to compute the exact posterior and compare it with the one approximated by DiBS. [...] I would like to see how the numbers from Table 4 change if one makes 3000 SVGD iterations. Does the uncertainty estimation remain the same?
>
> The question of optimally evaluating the posterior is commonly known to be challenging in Bayesian structure learning, because the true posterior is in general not available. Since we are aware of this difficulty, we report four different evaluation metrics throughout the experiment section, trying to capture as many relevant performance aspects of Bayesian Structure Learning as possible. For this, we rely on the available metrics of the existing literature [10, 21, 40, 54, 55]. In addition, our evaluation metrics also reflect quantities that are of practical interest in posterior inference used for downstream tasks, e.g., the I-MLL/I-LL in active causal discovery [12, 14]. The AUROC specifically evaluates proper uncertainty quantification on the edge level. In line with the existing literature, this is the most thorough evaluation methodology that is applicable to BN posteriors beyond 5 nodes. We are open to suggestions on how to further improve the evaluation of the posterior.
>
> Following your suggestion for the experiment of Appendix E.4., we ran DiBS for 3000 iterations instead of only 500. The results as shown in Table 4 are qualitatively unchanged because the particles converge early in this small setting. (For 3000 SVGD iterations, the values in Table 4  are unchanged at the given decimal precision).
>
> > L268: Is 30 samples from the posterior enough to estimate the various performance metrics? It would be interesting to see how much the metrics vary when one repeats the sampling. Do the performance metrics change a lot when using more samples from the posterior?
>
> We report the mean and box plots/standard deviations over 30 random repeats in all figures/tables to address the question of how much the metrics change when one repeats the sampling. When evaluating DiBS and the baseline methods, we have found that there is no significant difference in the evaluation metrics between using 30 samples to approximate the posterior and, e.g., 100 samples. This observation is consistent across all metrics (E-SHD, AUROC, etc.) and methods ($MC^3$, bootstrap, and DiBS variants). We would, however, be happy to increase the number of samples for the final version wherever computationally feasible, if you recommend that we do so.
>
> > In Section 4.3, titled “Black box bayesian inference”, I was not sure what the actual algorithm suggested was. I believe the title is a bit misleading since no specific inference procedure is suggested in this section. The first paragraph of the following section also implicitly mentions that some inference procedure was suggested above: “... translates learning discrete graph structures into an inference problem over the continuous variable Z.” [...] Please clarify.
>
> Following your suggestion that the subsection title was misleading, we changed the title of Section 4.3 to “Estimators for Gradient-Based Bayesian Inference”. In general, Section 4.3 covers the details on how to estimate the gradients of the posterior densities inferred in the DiBS framework. We do not propose an “inference method” or “algorithm” here, but merely derive the estimators to be able to apply existing gradient-based inference methods. Approximate Bayesian inference methods such as SVGD used here, but also e.g. Hamiltonian Monte Carlo [58] and Stochastic gradient Langevin dynamics [59] compute full posterior approximations (not MAP estimates) using score estimators as in Proposition 2 and Eqs. 12-14. We propose how this inference is actually performed later in Section 5. We hope that this clarifies the misunderstanding of the referenced sentence in lines 183-184, where we explicitly state that we formulate the task into an “inference problem over the continuous variable $Z$” (line 184), not a procedure.
>
> > Equation (15) felt a bit hard to interpret given that no explanation of the stein variational gradient descent was provided. Maybe adding some general explanation of this method in the background section would help, or maybe just refer to Algorithm 1 at the very beginning of Section 5.
>
> We agree that an additional background section on SVGD helps facilitate the overall understanding of our method in Section 5. Following your suggestion, we have added a whole additional background section on SVGD right before Appendix C and referenced it towards the beginning of Section 5.
>
> Finally, we address your minor comments:
>
> > Equation (8): Why using E_{G | Z}[h(G)] instead of h(G_\alpha(Z)) ? After all, h was designed to take as input weighted adjacency matrices.
>
> We have experimented with both but found that $E_{G | Z}[h(G)]$ performs better than $h(G_\alpha(Z))$.  The former formulation decouples $\alpha$ from the degree of relaxation in the adjacency matrix $G$. In other words, in $E_{G | Z}[h(G)]$, graph samples $G$ are discrete (or Gumbel relaxations with $\tau=1=\text{const.}$ in gradient estimation) and hence _independent of $\alpha$_ throughout the iterations of SVGD. By contrast, in $h(G_\alpha(Z))$, the matrix of probabilities $G_\alpha(Z)$ would be affected by $\alpha$ and suffer from a vanishing gradient especially initially when $\alpha$ is small.
>
> > Equation (17) if <u_i, v_j> = 0, the limit is ½ right? Probably not relevant since this concerns a set of measure zero.
>
> Correct, this case concerns a set of measure zero, so we use Eq. 17 as the natural definition for when $\alpha$ goes to infinity in practice.
>
> > For each method, which thresholds were explored to compute the AUROC, and how were they selected? Also regarding the AUROC, how is p(g_ij = 1 | D) estimated? I assume by counting the proportion of graphs with g_ij = 1 among the 30 samples from the posterior?
>
> In general, each method returns a posterior approximation of $p(G|D)$. Then, for each possible edge $(i,j)$, the marginal posterior edge probability is the posterior mean of an indicator for that edge, i.e. $p(g_{ij} = 1|D) = E_{G|D} I[g_{ij} = 1]$, which amounts to counting the proportion of graphs with $g_{ij} = 1$ (and weighted counting for the DAG bootstrap variants and DiBS+). The AUROC of every method is computed by viewing the presence of each possible edge $(i,j)$ as a binary classification task with predicted probability $p(g_{ij} = 1|D)$. All possible confidence thresholds between 0 and 1 (that lead to changes in classification decisions) are used to compute the ROC curve.
>
> In response to your question, we have added an additional subsection to Appendix D containing more detailed descriptions of the evaluation metrics, in particular AUROC and I-MLL/I-LL.
>
> > Confused by ijk in equation (8), I thought Z was a k x 2d matrix (line 129)
>
> We were viewing $Z$ as a $k \times d \times 2$ tensor. However, your comment tells us that viewing $Z$ as $k \times 2d$ is actually more clear. We have changed this detail in the manuscript.
>
> > How is E-SHD estimated? I am guessing via Monte Carlo estimation using the samples from the posterior. To clarify.
>
> Correct, the E-SHD is computed via Monte Carlo estimation using samples from the posterior, since E-SHD (Eq. 18) (and also neg. LL (Eq. 19)) are expectations under the posterior. Note that the DAG bootstrap variants and DiBS+ use the weighted mixture rather than the empirical distribution of samples.
>
>
> We hope that we have addressed your questions and concerns. Please do not hesitate to reach out for further clarifications if needed.

---

> > ### Author Response · Authors · 2021-08-25
> > **Follow-up**
> >
> > Dear Reviewer GxXY,
> >
> > We hope you had a chance to look at our point-by-point explanations, where we provide the specific clarifications you requested. Since you indicated that you are open to revise your score, we would like to make sure that we have addressed your concerns sufficiently and changed your assessment of our work for the better; should that not be the case, please do not hesitate to get in touch with us. Thank you again for your valuable feedback!

---

> > > ### Comment · Reviewer_GxXy · 2021-08-31
> > > **Score update**
> > >
> > > I thank the authors for their thorough response and I apologize for the late reply. I was satisfied with the response and overall I thought the paper was very well written. I found the extra experiment reassuring. Others on the decision committee agree with the authors regarding the fact that AUROC captures the quality of the uncertainty estimation. The authors understood my concerns and addressed them very clearly. Consequently, I am happy to raise my score from 6 to 7.

---

### Official Review · Reviewer_aXYX · 2021-07-16

**Rating:** 7
**Confidence:** 3

**Summary:**

This paper addresses the structure learning problem, in which the goal is to find a directed graphical model (structure and parameters) that accurately models an observed dataset. The authors work in the Bayesian structure learning paradigm, defining a prior distribution over graphical models and aiming to infer a posterior. Rather than defining a discrete prior over the unknown graph structure $G$, the authors define a prior over a continuous latent variable $Z$, and a conditional distribution on $G$ given $Z$, such that the data likelihood, $P(D \mid Z) = E_{G \mid Z}(P(D \mid G))$, is differentiable with respect to $Z$. This enables gradient-based inference methods to be applied to the problem of inferring $Z$, marginalizing $G$—if the necessary gradients can be efficiently estimated. In principle, any gradient-based inference method could be applied, but the authors spell out details of a Stein Variational Gradient Descent procedure. This proposed technique, and a variant in which an additional reweighting step is applied, are evaluated on synthetic data, against MCMC-based and non-parametric bootstrapping baselines.

**Limitations And Societal Impact:**

Broader impact is explicitly discussed (with a focus on the potential positives), but I don't see an explicit discussion of limitations. Some questions that might be worth considering:

* Is it really the case that other black-box inference methods would be easy to apply to your examples (L332)? For example, I don't believe Hamiltonian Monte Carlo can easily make use of the gradient estimators you give in Section 4.3.

* The score function estimator (Eq 14) is often described as having unacceptably high variance. You appear to have used it successfully in Section 6.2, but do you expect it will scale to e.g. models with parameters $\Theta$ that can't be analytically marginalized, or to higher-dimensional data?

**Main Review:**

This paper presents a novel approach to the important but difficult problem of Bayesian structure learning. Bayesian approaches to the problem are conceptually appealing (define a prior over graphs, and the obvious data likelihood), but hard to implement in practice, because they involve combinatorial search over a high-dimensional discrete space. This paper applies recent advances in differentiable characterizations of graph "acyclicness" to formulate a structure learning model to which (some) gradient-based inference methods can be applied.

Although various components of this approach have been previously proposed (e.g., structure learning as continuous optimization, and "inner product" priors for edge probabilities in graphs), this work presents a valuable synthesis of these ideas into a coherent model and inference algorithm for Bayesian structure learning. The empirical validation is thorough, and I appreciated the variety of different metrics used to evaluate performance.

One aspect I found unsatisfying was the move from DiBS to DiBS+. The unweighted particle collection is already meant to represent the posterior; wouldn't weighting it by the unnormalized target "double-count," further concentrating the inferred posterior? That this performs better than DiBS using the metrics investigated here suggests either that something is off about the metrics (i.e., they do not reward proper uncertainty?), or that the inference algorithm is under-confident / producing higher-uncertainty estimates than it should. Or am I misunderstanding something? One question for the authors is whether you have tried applying a similar re-weighting to samples from the MC^3 baselines you consider (weighting according to the posterior densities they target).

Other minor questions:

* In the BGe marginal likelihood, is the likelihood 0 for graphs G with cycles?
* Maybe I missed it, but could you report -log p(D^{test} | G*, Theta*) for the true data-generating graph? This may help calibrate the negative log likelihood numbers you report.
* In Section 7, where does "the consensus network" come from?

**Time Spent Reviewing:**

4.5 hours

---

> ### Author Response · Authors · 2021-08-09
> **Response to Reviewer aXYX**
>
> We thank you for the effort in reviewing our work and the valuable feedback. We would like to address your questions and concerns by starting with your mentioned discussion of explicit limitations:
>
> > Is it really the case that other black-box inference methods would be easy to apply to your examples (L332)
>
> Correct, any black-box inference method relying on the gradient of the target distribution can be applied on top of our framework. For example, directly analogous to SVGD used in this work, Hamiltonian Monte Carlo (HMC) [58] and Stochastic gradient Langevin dynamics (SGLD) [59] solely rely on the score $\nabla_x \log p(x)$ to approximate a target distribution $p$. For DiBS, the target distributions to be inferred are $p(Z|D)$ and $p(Z, \Theta | D)$, and estimators for the gradients of both of these densities are provided by Proposition 2  and Eqs. 12 and 14. Hence, there is no inherent difference between using DiBS with SVGD and, for example, HMC or SGLD, or other black-box inference methods relying on the score of the target density.
>
> > The score function estimator (Eq 14) is often described as having unacceptably high variance. You appear to have used it successfully in Section 6.2, but do you expect it will scale to e.g. models with parameters that can't be analytically marginalized, or to higher-dimensional data?
>
> In the joint inference setting, where the parameters are not marginalized out, the score function estimator (Eq. 14) would only be required for $\nabla_Z \log p(Z, \Theta | D)$ and not for $\nabla_\Theta \log p(Z, \Theta | D)$ (c.f. Eq. 10 and 11). For this reason, we do not expect models in which parameters cannot be analytically marginalized to affect the performance of the score function estimator. We show in Figure 5 of Appendix D.5 that the estimator’s variance can indeed be problematic in higher-dimensional settings ($d=50$), so we opt to use the Gumbel-softmax estimator (Eq. 12) whenever applicable (c.f. lines 193-196; contrast with Figures 6 and 7).
>
> > One aspect I found unsatisfying was the move from DiBS to DiBS+. The unweighted particle collection is already meant to represent the posterior; wouldn't weighting it by the unnormalized target "double-count," further concentrating the inferred posterior? [...]
>
> Overall, we motivate weighting each inferred particle by its unnormalized posterior, i.e., moving from DiBS to DiBS+, by the equivalent weighting procedure performed by the nonparametric DAG bootstrap [41]. In addition, note that while SVGD operates in the continuous space of $Z^{(m)}$ to approximate the posterior _density_ $p(Z|D)$, DiBS+ ultimately weighs the _discrete_ adjacency matrices $G^{(m)}$ to approximate the posterior $p(G|D)$, i.e., in the realm of probability mass functions rather than densities, beyond the density inferred by SVGD itself. However, we readily admit that we do not have a strong theoretical justification for the weighting at this point, and we will be happy to flag this in the paper.
> We hypothesize that the weighting of DiBS+ helps empirically because most of the particles in the empirical distribution of inferred graph particles are unique (which is a consequence of the vast, super-exponentially large space of DAGs). As a result, the DiBS distribution is nearly uniform over a few “likely” DAG candidates, and additionally weighting these by their likelihood may result in a closer approximation of the true posterior.
>
> Finally, we address the minor comments:
>
> > In the BGe marginal likelihood, is the likelihood 0 for graphs G with cycles?
>
> In general, the BGe marginal is properly defined also for cyclic graphs as it factors into node-wise family scores [17, 18]. Thus, it usually gives a likelihood > 0 for cyclic graphs. What prevents us from inferring cyclic graphs is the prior $p_\beta(Z)$ in Eq. 8. As we increase $\beta$ during inference, our prior assigns almost zero probability to any $Z$ modeling a cyclic graph.
>
> > Maybe I missed it, but could you report -log p(D^{test} | G*, Theta*) for the true data-generating graph?
>
> In the marginal inference setting for 20-node BNs (Figure 2), the neg. MLL (neg. I-MLL) achieved by the ground truth BN $(G^*, \Theta^*)$ is 745.2 $\pm$ 44.7 (569.1 $\pm$ 22.4). In the joint inference settings for linear Gaussian BNs (Figure 3), the neg. LL (neg. I-LL) achieved by the ground truth BN $(G^*, \Theta^*)$ is 539.7 $\pm$ 27.1 (429.8 $\pm$ 9.8). In the nonlinear case (Figure 4), the neg. LL (neg. I-LL) is 9,495.3 $\pm$ 7,163.8 (7,701.9 $\pm$ 6,017.3).
>
> For 50-node BNs (Appendix, Figures 5, 6, and 7), in the same order as above, the metrics achieved by the ground truth BNs are neg. MLL (neg. I-MLL) of 1,850.4 $\pm$ 81.0 (1,421.4 $\pm$ 33.6), and  neg. LL (neg. I-LL) of 1,329.6 $\pm$ 60.2 (1,068.2 $\pm$ 16.8) and 22,424.6 $\pm$ 11,931.6 (17,694.7 $\pm$ 10,280.3), respectively.
>
> All of these values are the mean and standard deviation across the 30 random BN instances  used to evaluate each setting and correspond to the Erdos-Renyi graph setting (i.e., top rows of figures). The results for scale-free graphs (i.e., bottom rows of figures) are qualitatively similar.
>
> > In Section 7, where does "the consensus network" come from?
>
> The consensus network underlying the dataset by Sachs et al. in Section 7 was established via a sequence of lab experiments performed in studies of this protein signaling network and is generally accepted by the community [3].
>
> We hope that we were able to address your questions. Please reach out if further clarification is needed.

---

> > ### Comment · Reviewer_aXYX · 2021-09-01
> > **Thank you!**
> >
> > Thank you to the authors for this detailed response! I am maintaining my recommendation for acceptance.
> >
> > A couple brief comments in response:
> >
> > > Hamiltonian Monte Carlo (HMC) [58] and Stochastic gradient Langevin dynamics (SGLD) [59] solely rely on the score ∇_x log p(x)
> >
> > Yes, but I believe problems can arise when naively using estimates of gradients in HMC (i.e., instead of using exact gradients). See e.g. https://arxiv.org/abs/1402.4102.
> >
> > > We hypothesize that the weighting of DiBS+ helps empirically because most of the particles in the empirical distribution of inferred graph particles are unique (which is a consequence of the vast, super-exponentially large space of DAGs). As a result, the DiBS distribution is nearly uniform over a few “likely” DAG candidates, and additionally weighting these by their likelihood may result in a closer approximation of the true posterior.
> >
> > Interesting, thanks. It would be interesting to test this explanation... e.g., one could check to see if similar results can be achieved by running DiBS many times on the same problem, and averaging the resulting empirical distributions. Or perhaps there are other tests. It seems useful to figure out whether the DiBS method itself (without additional weighting) is capable of approximating the posterior well, or if it should be regarded as a heuristic method for finding 'likely graphs.'

---

### Official Review · Reviewer_voY1 · 2021-07-18

**Rating:** 8
**Confidence:** 4

**Summary:**

This paper proposes a fully differentiable (i.e., continuous-space) method for Bayesian learning of Bayesian network structures as well as parameters. The proposed method (DiBS) is compared against several existing state-of-the-art methods on simulated and real data sets, and the results demonstrate very competitive performance.

**Limitations And Societal Impact:**

Yes

**Main Review:**

This paper proposes a fully differentiable (i.e., continuous-space) method for Bayesian learning of Bayesian network structures as well as parameters. Authors propose a seemingly simple generative model (Fig. 1): a generative graph model is motivated by variational graph autoencoders and utilizes recent results on continuous characterization of acyclic graphs that are incorporated as priors in the generative model. Model inference is implemented with the Stein variational gradient descent, with the necessary technical terms being derived in the manuscript/appendix. The proposed method (DiBS) is compared against several existing state-of-the-art methods on simulated and real data sets, and the results demonstrate very competitive performance.

The manuscript is very clear and easy to read. To the best of my knowledge, this manuscript proposes a novel technique for structure/parameter learning for BNs that goes beyond the previous methods; it could even be fair to say that perhaps this manuscript opens a new direction for Bayesian learning for BN structure/parameters and for causal inference in general. Having worked on the BN learning in the past, I must say that I read the manuscript with great interest.  I have no real concerns about this work: all technical details seem correct, and simulations and results support the claims made in the manuscript.  I list some comments/suggestions and minor comments below that the authors can implement when revising their manuscript.

1. Overall the literature review appears to be very comprehensive. While it is true that most of the MCMC techniques for BN structure inference assume a closed form for the marginal likelihood (which makes it easy to implement e.g. Metropolis-Hasting algorithm over structures), previous methods also include reversible jump MCMC methods for inferring both the structure and parameters (proposed at least in the context of dynamics BNs).

2. Although Fig. 2 does not contain the most impressive results, I found it very convincing that the proposed method performs comparably with (or in terms of neg. (I-)MLL measures even slightly better than) the MC^3 for linear Gaussian BNs.

3. Page 7, row 268: why did you use so few samples (S=30)?

4. Appendix D.5, results for d=50: can you discuss about possible reasons why the method performs worse for the case where closed form for the MLL is available?

5. Sec. 7, phosphoproteomics data analysis: I would recommend authors to consider using neg. log.likel. measures on hold-out data in addition to comparing against the "ground truth network structure", as the ground truth may not be entirely correct (e.g. missing links).


**Time Spent Reviewing:**

3

---

> ### Author Response · Authors · 2021-08-09
> **Response to Reviewer voY1**
>
> We thank you for the careful assessment of our work. In the following, we provide a point-by-point reply to each of your minor comments and suggestions:
>
> > [...] previous methods also include reversible jump MCMC methods for inferring both the structure and parameters (proposed at least in the context of dynamics BNs).
>
> To our knowledge, in the context of BN structure learning, reversible jump MCMC is mostly applied to learn changes in structure in _non-stationary_ dynamic BNs, e.g. in Robinson and Hartemink’s “Non-stationary dynamic Bayesian networks” (2008) and Grzegorczyk and Husmeier’s “Non-stationary continuous dynamic Bayesian networks” (2009), and not to learn the joint distribution of graphs and parameters. These works do use the marginal likelihood. If you have concrete other work in mind that uses reversible jump MCMC for inferring both the structure and parameters, we would be happy to include them in our related work section.
>
> > Page 7, row 268: why did you use so few samples (S=30)?
>
> When evaluating DiBS and the baseline methods, we have found that there is no significant difference in the evaluation metrics between using 30 samples to approximate the posterior and, e.g., 100 samples. This observation is consistent across all metrics (E-SHD, AUROC, etc.) and methods ($MC^3$, bootstrap, and DiBS variants). We did not see a compelling reason to spend further compute resources here. We would, however, be happy to increase the number of samples for the final version wherever computationally feasible, if you recommend that we do so.
>
> > Appendix D.5, results for d=50: can you discuss about possible reasons why the method performs worse for the case where closed form for the MLL is available?
>
> As explained in lines 258-261, when using the closed-form BGe marginal likelihood to infer $p(G|D)$, DiBS cannot employ the Gumbel-softmax estimator and uses the score-function estimator instead, because the gradient with respect to the adjacency matrix is ill-defined in the BGe computation. As we point out in the caption of Figure 5 of Appendix D.5, we believe that the high variance of the score function estimator used here, in combination with this high-dimensional setting of 50 variables, is the main reason why the performance is worse compared to the joint inference case (where the Gumbel-softmax estimator can be used; see Figures 6 and 7 and lines 272-274). Note that inferring the 50-node posterior implies an inference task over the latent tensor $Z$ of $50 \cdot 50 \cdot 2 = 5,000$ dimensions. We expect that using more than our default 128 Monte Carlo samples for estimating the expectations (Eqs. 9 and 14) would remedy this.
>
> > Sec. 7, phosphoproteomics data analysis: I would recommend authors to consider using neg. log.likel. measures on hold-out data in addition to comparing against the "ground truth network structure", as the ground truth may not be entirely correct (e.g. missing links).
>
> We had previously already evaluated the held-out (marginal) log likelihoods for the dataset of Sachs et al. We found, however, that all methods performed fairly similarly, thus the results were inconclusive and so we chose not to report them. We hypothesize that this is due to high observation noise in this data set, making it challenging to infer exact predictions beyond the graph structure of dependencies.

---

> > ### Comment · Reviewer_voY1 · 2021-09-02
> > **Thanks**
> >
> > I will keep my original score (increasing the score wouId not be unfair either) and I strongly support that this manuscript will be accepted.

---

### Author Response · Authors · 2021-08-09
**General remark to all Reviewers**

We would like to thank all of the reviewers for their careful reading, assessment, and feedback. We are pleased that the reviewers recognize our contribution as opening “a new direction for Bayesian learning for BN structure/parameters and for causal inference in general” [voY1], that our “empirical validation is thorough” [aXYX], and that our method is “relevant for a lot of real-world applications and [...] a potentially valuable tool for practitioners” [ZLxN].  We have incorporated the reviewers’ individual suggestions in our manuscript. We address the individual questions and suggestions in separate responses.

---

### Decision · Program_Chairs · 2021-09-27

**Decision:**

Accept (Spotlight)

**Comment:**

Reviewers agree that the proposed use of continuous embeddings for Bayesian learning of directed graphical models is innovative and potentially high-impact.  Reviews generally have very positive comments about the manuscript's clear presentation and promising experimental results.  In future revisions, please do better discuss the motivations for the "DiBS+" variant, and clarify the points about technical details and evaluation metrics that were discussed with reviewers.